# A Fisher-Rao gradient flow for entropic mean-field min-max games

**Razvan-Andrei Lascu**                                                    *rl2029@hw.ac.uk*
*School of Mathematical and Computer Sciences, Heriot-Watt University, Edinburgh, UK, and Maxwell Institute for Mathematical Sciences, Edinburgh, UK*

**Mateusz B. Majka**                                                      *m.majka@hw.ac.uk*
*School of Mathematical and Computer Sciences, Heriot-Watt University, Edinburgh, UK, and Maxwell Institute for Mathematical Sciences, Edinburgh, UK*

**Łukasz Szpruch**                                                       *l.szpruch@ed.ac.uk*
*School of Mathematics, University of Edinburgh, UK, and The Alan Turing Institute, UK and Simtopia, UK*

**Reviewed on OpenReview:** *https://openreview.net/forum?id=Afc2CucRaR*

## Abstract

Gradient flows play a substantial role in addressing many machine learning problems. We examine the convergence in continuous-time of a *Fisher-Rao* (Mean-Field Birth-Death) gradient flow in the context of solving convex-concave min-max games with entropy regularization. We propose appropriate Lyapunov functions to demonstrate convergence with explicit rates to the unique mixed Nash equilibrium.

## 1 Introduction

The rapid progress of machine learning (ML) techniques such as Generative Adversarial Networks (GANs) (Goodfellow et al., 2014), adversarial learning (Madry et al., 2018), multi-agent reinforcement learning (Zhang et al., 2021) has propelled a surge of interest in the study of optimization problems on the space of probability measures in recent years. Particularly noteworthy are the numerous works, e.g., (Hsieh et al., 2019; Domingo-Enrich et al., 2020; Wang & Chizat, 2023; Lu, 2023; Trillos & Trillos, 2023; Kim et al., 2024), illustrating how training GANs and addressing adversarial robustness can be cast as min-max games over probability measures. In this setting, understanding the time evolution of the players' initial strategies to the equilibrium of the game leverages the use of gradient flows on the space of probability measures. A discussion about the applications of gradient flows in optimization and sampling can be found in a recent survey (Trillos et al., 2023).

The Fisher-Rao (FR) gradient flow has been recently studied in the context of mean-field optimization problems (Liu et al., 2023), accelerating Langevin-based sampling from multi-modal distributions (Lu et al., 2019; 2023), and training shallow neural networks in the mean-field regime (Rotskoff et al., 2019). The motivation for employing FR dynamics in sampling from multi-modal distributions is their ability to globally transport the mass of a probability density between modes without traversing low-probability regions (Lu et al., 2019). In the context of training neural networks, the benefit of using FR dynamics is similar in that they have the potential capability to avoid local minima of the loss function (Rotskoff et al., 2019).

The present paper aims to extend these results and focuses on the continuous-time convergence of the *Fisher-Rao* (FR) gradient flow to the unique mixed Nash equilibrium of an entropy-regularized min-max game.

## 1.1 Notation and setup

For any $\mathcal{Z} \subseteq \mathbb{R}^d$, we denote by $\mathcal{P}_{\mathrm{ac}}(\mathcal{Z})$ the space of probability measures on $\mathcal{Z}$ which are absolutely continuous with respect to the Lebesgue measure. Following a standard convention, elements in $\mathcal{P}_{\mathrm{ac}}(\mathcal{Z})$ denote probability measures as well as their densities. Let $\mathcal{X}, \mathcal{Y} \subseteq \mathbb{R}^d$ and fix two reference probability measures $\pi(\mathrm{d}x) \propto e^{-U^\pi(x)}\mathrm{d}x \in \mathcal{P}_{\mathrm{ac}}(\mathcal{X})$ and $\rho(\mathrm{d}y) \propto e^{-U^\rho(y)}\mathrm{d}y \in \mathcal{P}_{\mathrm{ac}}(\mathcal{Y})$, where $U^\pi : \mathcal{X} \to \mathbb{R}$ and $U^\rho : \mathcal{Y} \to \mathbb{R}$ are two measurable functions. The relative entropy $\mathrm{D}_{\mathrm{KL}}(\cdot|\pi) : \mathcal{P}(\mathcal{X}) \to [0, \infty]$ with respect to $\pi$ is given by

$$\mathrm{D}_{\mathrm{KL}}(\nu|\pi) = \begin{cases} \int_{\mathcal{X}} \log\left(\frac{\nu(x)}{\pi(x)}\right)\nu(x)\mathrm{d}x, & \text{if } \nu \text{ is absolutely continuous with respect to } \pi, \\ +\infty, & \text{otherwise,} \end{cases}$$

and we define $\mathrm{D}_{\mathrm{KL}}(\mu|\rho)$ analogously for any $\mu \in \mathcal{P}(\mathcal{Y})$. Let $F : \mathcal{P}(\mathcal{X}) \times \mathcal{P}(\mathcal{Y}) \to \mathbb{R}$ be a convex-concave (possibly non-linear) function and $\sigma > 0$ be a regularization parameter. We consider the min-max problem

$$\min_{\nu \in \mathcal{P}(\mathcal{X})} \max_{\mu \in \mathcal{P}(\mathcal{Y})} V^\sigma(\nu, \mu), \text{ with } V^\sigma(\nu, \mu) := F(\nu, \mu) + \frac{\sigma^2}{2}\left(\mathrm{D}_{\mathrm{KL}}(\nu|\pi) - \mathrm{D}_{\mathrm{KL}}(\mu|\rho)\right). \tag{1}$$

In order to solve (1), one typically aims to identify *mixed Nash equilibria* (MNEs) (von Neumann et al., 1944; Nash, 1951), characterized by pairs of measures $(\nu_\sigma^*, \mu_\sigma^*) \in \mathcal{P}(\mathcal{X}) \times \mathcal{P}(\mathcal{Y})$ that satisfy

$$V^\sigma(\nu_\sigma^*, \mu) \leq V^\sigma(\nu_\sigma^*, \mu_\sigma^*) \leq V^\sigma(\nu, \mu_\sigma^*), \quad \text{for all } (\nu, \mu) \in \mathcal{P}(\mathcal{X}) \times \mathcal{P}(\mathcal{Y}). \tag{2}$$

It is important to highlight that when $F$ is bilinear and $\sigma = 0$, i.e., $V^0(\nu, \mu) = \int_{\mathcal{Y}} \int_{\mathcal{X}} f(x, y)\nu(\mathrm{d}x)\mu(\mathrm{d}y)$, for some measurable function $f : \mathcal{X} \times \mathcal{Y} \to \mathbb{R}$, measures characterized by (2) represent MNEs in the classical sense of a two-player zero-sum game. Results concerned with the existence and uniqueness of an MNE for (1) are presented in Appendix A in Lascu et al. (2023) (alternatively, see Appendix C.2, C.3 in Domingo-Enrich et al. (2020) for the case when $F$ is bilinear). It is also proved in Lascu et al. (2023) that $V^\sigma$ $\Gamma$-converges to $F$ as $\sigma \downarrow 0$, under mild assumptions on $F, \pi$ and $\rho$.

In optimization, the monotonic decrease of the objective function along the gradient flow is key to proving convergence. However, for min-max games the monotonic decrease no longer holds due to the conflicting actions of the players. Hence, a suitable Lyapunov function is needed. A common choice is the so-called Nikaidò-Isoda (NI) error (Nikaidô & Isoda, 1955), which, for all $(\nu, \mu) \in \mathcal{P}(\mathcal{X}) \times \mathcal{P}(\mathcal{Y})$, can be defined as

$$\mathrm{NI}(\nu, \mu) := \max_{\mu' \in \mathcal{P}(\mathcal{Y})} V^\sigma(\nu, \mu') - \min_{\nu' \in \mathcal{P}(\mathcal{X})} V^\sigma(\nu', \mu).$$

From the saddle point condition (2), it follows that $\mathrm{NI}(\nu, \mu) \geq 0$ and $\mathrm{NI}(\nu, \mu) = 0$ if and only if $(\nu, \mu)$ is a MNE. We will also consider an alternative Lyapunov function given by (6).

In what follows, we will introduce the FR gradient flow on the space $(\mathcal{P}_{\mathrm{ac}}(\mathcal{X}) \times \mathcal{P}_{\mathrm{ac}}(\mathcal{Y}), \mathrm{FR})$, where FR is the Fisher-Rao distance defined by (3).

## 1.2 Fisher-Rao (mean-field birth-death) gradient flow

The Fisher-Rao metric was introduced by Rao (1992) via the Fisher information matrix, and since then has been extensively studied in the context of information geometry (Amari, 2016; Ay et al., 2017). In this work, we are mainly focusing on the the dynamical formulation of the Fisher-Rao metric (see e.g. Section 2.2 in Gallouët & Monsaingeon (2017), Appendix C.2.1 in Yan et al. (2023) and Section 3.3 in Kondratyev & Vorotnikov (2019)). For any $\lambda_0, \lambda_1 \in \mathcal{P}_{\mathrm{ac}}(\mathcal{M})$, with $\mathcal{M} \subseteq \mathbb{R}^d$, the variational representation of the FR distance is given by

$$\mathrm{FR}^2(\lambda_0, \lambda_1) := \inf\left\{\int_0^1 \int_{\mathcal{M}} \left|r_s(x) - \int_{\mathcal{M}} r_s(y)\lambda_s(\mathrm{d}y)\right|^2 \lambda_s(\mathrm{d}x)\mathrm{d}s : \partial_t \lambda_t = \lambda_t\left(r_t - \int_{\mathcal{M}} r_t(y)\lambda_t(\mathrm{d}y)\right)\right\}, \tag{3}$$

where the infimum is taken over all curves $[0, 1] \ni t \mapsto (\lambda_t, r_t) \in \mathcal{P}_{\mathrm{ac}}(\mathcal{M}) \times L^2(\mathcal{M}; \lambda_t)$ solving

$$\partial_t \lambda_t = \lambda_t\left(r_t - \int_{\mathcal{M}} r_t(y)\lambda_t(\mathrm{d}y)\right) \tag{4}$$

in the distributional sense, such that $t \mapsto \lambda_t$ is weakly continuous with endpoints $\lambda_0$ and $\lambda_1$. The reaction term $r_t(x) \in \mathbb{R}$ is a scalar that dictates how much mass is created/destroyed at time $t > 0$ and position $x \in \mathbb{R}$. The integral in (4) guarantees that $\lambda_t$ is a probability measure for all $t \geq 0$.

Inspired by Liu et al. (2023), we consider the Fisher-Rao (mean-field birth-death) gradient flow on the space $(\mathcal{P}_{\mathrm{ac}}(\mathcal{X}) \times \mathcal{P}_{\mathrm{ac}}(\mathcal{Y}), \mathrm{FR})$ in the setting of (1). As opposed to the mean-field Best Response dynamics studied in Lascu et al. (2023), which is another flow that can be used to solve (1), and which relies on introducing a fixed point perspective on min-max games, the FR dynamics utilize a gradient flow $(\nu_t, \mu_t)_{t \geq 0}$ in the Fisher-Rao geometry. As a first attempt at defining a Fisher-Rao gradient flow for solving (1), consider

$$\begin{cases} \partial_t \nu_t(x) = -\frac{\delta V^\sigma}{\delta \nu}(\nu_t, \mu_t, x)\nu_t(x), \\ \partial_t \mu_t(y) = \frac{\delta V^\sigma}{\delta \mu}(\nu_t, \mu_t, y)\mu_t(y), \end{cases} \tag{5}$$

with initial condition $(\nu_0, \mu_0) \in \mathcal{P}_{\mathrm{ac}}(\mathcal{X}) \times \mathcal{P}_{\mathrm{ac}}(\mathcal{Y})$. We adopt the convention that

$$\int_{\mathcal{X}} \frac{\delta V^\sigma}{\delta \nu}(\nu_t, \mu_t, x)\nu_t(\mathrm{d}x) = \int_{\mathcal{Y}} \frac{\delta V^\sigma}{\delta \mu}(\nu_t, \mu_t, y)\mu_t(\mathrm{d}y) = 0$$

since the flat derivatives of $V^\sigma$ are uniquely defined up to an additive shift (see Definition B.1 in Section B). Thus, the total mass 1 of probability measures is still preserved along the gradient flow. A formal derivation of the Fisher-Rao gradient flow can be found in Appendix C.

**Remark 1.1.** *Suppose $V^0$ is a bilinear payoff function and any mixed strategies $(\nu, \mu)$ are supported on finite sets of $m$ and $n$ pure strategies $\mathcal{X} \coloneqq \{x_1, x_2, \cdots, x_m\}$ and $\mathcal{Y} \coloneqq \{y_1, y_2, \cdots, y_n\}$, respectively. Then $\nu \in \mathcal{P}(\mathcal{X})$ and $\mu \in \mathcal{P}(\mathcal{Y})$ can be approximated by the empirical measures $\nu^m \coloneqq \frac{1}{m}\sum_{i=1}^m \delta_{x_i}$ and $\mu^n \coloneqq \frac{1}{n}\sum_{j=1}^n \delta_{y_j}$, respectively, and hence the Fisher-Rao gradient flow (5) retrieves the replicator dynamics studied in evolutionary game theory (Cressman & Tao, 2014; Hofbauer & Sigmund, 1998), see also (Abe et al., 2022) for a related replicator-mutant dynamics.*

### 1.2.1 Sketch of convergence proof for the FR gradient flow

For the sake of presenting an intuitive heuristic argument, we ignore here for now that the *flat derivatives* $(\nu, \mu, x) \mapsto \frac{\delta V^\sigma}{\delta \nu}(\nu, \mu, x)$ and $(\nu, \mu, y) \mapsto \frac{\delta V^\sigma}{\delta \mu}(\nu, \mu, y)$ may not exist for the $V^\sigma$ defined in (1), due to the relative entropy term $\mathrm{D}_{\mathrm{KL}}$ being only lower semicontinuous with respect to the weak convergence topology (for this reason, in our analysis in Section 2, we will replace these two derivatives with appropriately defined auxiliary functions $a$ and $b$). Nevertheless, we will now demonstrate that choosing the flow $(\nu_t, \mu_t)_{t \geq 0}$ as in (5), makes the function

$$t \mapsto \mathrm{D}_{\mathrm{KL}}(\nu_\sigma^*|\nu_t) + \mathrm{D}_{\mathrm{KL}}(\mu_\sigma^*|\mu_t) \tag{6}$$

decrease in $t$, under the assumption of convexity-concavity of $F$. Let $(\nu, \mu) \in \mathcal{P}_{\mathrm{ac}}(\mathcal{X}) \times \mathcal{P}_{\mathrm{ac}}(\mathcal{Y})$. Then assuming the existence of the flow $(\nu_t, \mu_t)_{t \geq 0}$ satisfying (5), and the differentiablity of the map $t \mapsto \mathrm{D}_{\mathrm{KL}}(\nu|\nu_t) + \mathrm{D}_{\mathrm{KL}}(\mu|\mu_t)$, we formally have that

$$\begin{aligned} \frac{\mathrm{d}}{\mathrm{d}t}\left(\mathrm{D}_{\mathrm{KL}}(\nu|\nu_t) + \mathrm{D}_{\mathrm{KL}}(\mu|\mu_t)\right) &= \int_{\mathcal{X}} \partial_t\left(\nu(x)\log\frac{\nu(x)}{\nu_t(x)}\right)\mathrm{d}x + \int_{\mathcal{Y}} \partial_t\left(\mu(y)\log\frac{\mu(y)}{\mu_t(y)}\right)\mathrm{d}y \\ &= -\int_{\mathcal{X}} (\nu(x) - \nu_t(x))\frac{\partial_t\nu_t(x)}{\nu_t(x)}\mathrm{d}x - \int_{\mathcal{Y}} (\mu(y) - \mu_t(y))\frac{\partial_t\mu_t(y)}{\mu_t(y)}\mathrm{d}y \\ &= \int_{\mathcal{X}} \frac{\delta V^\sigma}{\delta \nu}(\nu_t, \mu_t, x)(\nu - \nu_t)(\mathrm{d}x) - \int_{\mathcal{Y}} \frac{\delta V^\sigma}{\delta \mu}(\nu_t, \mu_t, y)(\mu - \mu_t)(\mathrm{d}y), \end{aligned}$$

where the second equality follows from the fact that $\int \partial_t\nu_t(x)\mathrm{d}x = \int \partial_t\mu_t(y)\mathrm{d}y = 0$. Then, assuming that $\nu \mapsto F(\nu, \mu)$ and $\mu \mapsto F(\nu, \mu)$ are convex and concave, respectively (see Assumption 1), we observe that $\nu \mapsto V^\sigma(\nu, \mu)$ and $\mu \mapsto V^\sigma(\nu, \mu)$ are $\sigma$-strongly-convex and $\sigma$-strongly-concave relative to $\mathrm{D}_{\mathrm{KL}}$, respectively (see Lemma 3.1 in Section A), that is

$$V^\sigma(\nu, \mu_t) - V^\sigma(\nu_t, \mu_t) \geq \int_{\mathcal{X}} \frac{\delta V^\sigma}{\delta \nu}(\nu_t, \mu_t, x)(\nu - \nu_t)(\mathrm{d}x) + \frac{\sigma^2}{2}\mathrm{D}_{\mathrm{KL}}(\nu|\nu_t),$$

$$V^\sigma(\nu_t, \mu) - V^\sigma(\nu_t, \mu_t) \leq \int_{\mathcal{Y}} \frac{\delta V^\sigma}{\delta \mu}(\nu_t, \mu_t, y)(\mu - \mu_t)(\mathrm{d}y) - \frac{\sigma^2}{2} \mathrm{D_{KL}}(\mu|\mu_t).$$

Therefore, we obtain that

$$\frac{\mathrm{d}}{\mathrm{d}t} \left( \mathrm{D_{KL}}(\nu|\nu_t) + \mathrm{D_{KL}}(\mu|\mu_t) \right) \leq V^\sigma(\nu, \mu_t) - V^\sigma(\nu_t, \mu_t) + V^\sigma(\nu_t, \mu_t) - V^\sigma(\nu_t, \mu)$$
$$- \frac{\sigma^2}{2} \mathrm{D_{KL}}(\nu|\nu_t) - \frac{\sigma^2}{2} \mathrm{D_{KL}}(\mu|\mu_t). \quad (7)$$

Setting $(\nu, \mu) = (\nu_\sigma^*, \mu_\sigma^*)$ in (7), using the the saddle point condition (2), i.e., $V^\sigma(\nu_\sigma^*, \mu_t) - V^\sigma(\nu_t, \mu_\sigma^*) \leq 0$, and applying Gronwall's inequality gives

$$\mathrm{D_{KL}}(\nu_\sigma^*|\nu_t) + \mathrm{D_{KL}}(\mu_\sigma^*|\mu_t) \leq e^{-\frac{\sigma^2}{2}t} \left( \mathrm{D_{KL}}(\nu_\sigma^*|\nu_0) + \mathrm{D_{KL}}(\mu_\sigma^*|\mu_0) \right). \quad (8)$$

Since $\mathrm{D_{KL}}(\nu_\sigma^*|\nu) + \mathrm{D_{KL}}(\mu_\sigma^*|\mu) \geq 0$ with equality if and only if $\nu = \nu_\sigma^*$ and $\mu = \mu_\sigma^*$, it follows that the unique MNE of (1) is achieved with exponential rate $e^{-\frac{\sigma^2}{2}t}$.

To prove convergence in terms of the NI error, first observe that, for any $(\nu, \mu) \in \mathcal{P}(\mathcal{X}) \times \mathcal{P}(\mathcal{Y})$, we can write

$$V^\sigma(\nu_t, \mu) - V^\sigma(\nu, \mu_t) = V^\sigma(\nu_t, \mu) - V^\sigma(\nu_\sigma^*, \mu) + V^\sigma(\nu_\sigma^*, \mu) - V^\sigma(\nu, \mu_\sigma^*) + V^\sigma(\nu, \mu_\sigma^*) - V^\sigma(\nu, \mu_t).$$

By Theorem 2.2 and Assumption 2, there exist $C_{1,\sigma}, C_{2,\sigma} > 0$ depending on $\sigma$ such that

$$\left| \frac{\delta V^\sigma}{\delta \nu}(\nu_t, \mu, x) \right| \leq C_{1,\sigma}, \quad \left| \frac{\delta V^\sigma}{\delta \mu}(\nu, \mu_t, y) \right| \leq C_{2,\sigma},$$

for all $(\nu, \mu) \in \mathcal{P}(\mathcal{X}) \times \mathcal{P}(\mathcal{Y})$, and all $(x, y) \in \mathcal{X} \times \mathcal{Y}$. Then, by Lemma 3.1 and Pinsker's inequality, we can show that

$$V^\sigma(\nu_t, \mu) - V^\sigma(\nu_\sigma^*, \mu) + V^\sigma(\nu, \mu_\sigma^*) - V^\sigma(\nu, \mu_t) \leq 2C_\sigma \sqrt{\mathrm{D_{KL}}(\nu_\sigma^*|\nu_t) + \mathrm{D_{KL}}(\mu_\sigma^*|\mu_t)},$$

where $C_\sigma := \max\{C_{1,\sigma}, C_{2,\sigma}\}$. Hence, by (8), we obtain

$$V^\sigma(\nu_t, \mu) - V^\sigma(\nu, \mu_t) \leq 2C_\sigma e^{-\frac{\sigma^2}{4}t} \sqrt{\mathrm{D_{KL}}(\nu_\sigma^*|\nu_0) + \mathrm{D_{KL}}(\mu_\sigma^*|\mu_0)} + V^\sigma(\nu_\sigma^*, \mu) - V^\sigma(\nu, \mu_\sigma^*). \quad (9)$$

Since $(\nu_\sigma^*, \mu_\sigma^*)$ is the MNE of (1), we have

$$\max_\mu V^\sigma(\nu_\sigma^*, \mu) = \min_\nu V^\sigma(\nu, \mu_\sigma^*) = V^\sigma(\nu_\sigma^*, \mu_\sigma^*).$$

Therefore, maximizing over $(\nu, \mu) \in \mathcal{P}(\mathcal{X}) \times \mathcal{P}(\mathcal{Y})$ in (9) gives

$$\mathrm{NI}(\nu_t, \mu_t) \leq 2C_\sigma e^{-\frac{\sigma^2}{4}t} \sqrt{\mathrm{D_{KL}}(\nu_\sigma^*|\nu_0) + \mathrm{D_{KL}}(\mu_\sigma^*|\mu_0)}.$$

Finally, observe that, by (18), we have

$$\mathrm{D_{KL}}(\nu_\sigma^*|\nu_0) + \mathrm{D_{KL}}(\mu_\sigma^*|\mu_0) < \infty.$$

## 1.3 Our contribution

We prove the existence of the FR gradient flow $(\nu_t, \mu_t)_{t \geq 0}$ and show that it converges with rates $\mathcal{O}\left(e^{-\frac{\sigma^2}{2}t}\right)$ and $\mathcal{O}\left(e^{-\frac{\sigma^2}{4}t}\right)$ to the unique MNE of (1) with respect to the Lyapunov functions $t \mapsto \mathrm{D_{KL}}(\nu_\sigma^*|\nu_t) + \mathrm{D_{KL}}(\mu_\sigma^*|\mu_t)$ and $t \mapsto \mathrm{NI}(\nu_t, \mu_t)$.

### 1.4 Related works

Recent intensive research has been dedicated to examining the convergence of the Wasserstein gradient flow to MNEs within the specific formulation of game (1) in which $F$ is bilinear ($F(\nu, \mu) = \int_{\mathcal{Y}} \int_{\mathcal{X}} f(x, y)\nu(\mathrm{d}x)\mu(\mathrm{d}y)$) and regularized by entropy rather than relative entropy $\mathrm{D}_{\mathrm{KL}}$. The spaces $\mathcal{X}$ and $\mathcal{Y}$ are assumed to be either compact smooth manifolds without boundary, embedded in the Euclidean space, or Euclidean tori, while $f$ exhibits sufficient regularity, typically being at least continuously differentiable with Lipschitz conditions satisfied by $\nabla_x f$ and $\nabla_y f$. This line of research is explored in works such as Domingo-Enrich et al. (2020); Ma & Ying (2022); Lu (2023); Wang & Chizat (2023).

In this context, Ma & Ying (2022); Lu (2023) investigate the convergence of the Wasserstein gradient flow, proving exponential convergence to the MNE when the players' flows $(\nu_t)_{t \geq 0}$ and $(\mu_t)_{t \geq 0}$ converge at different rates. In Ma & Ying (2022), the analysis considers the scenario where one player's flow reaches equilibrium while the other remains governed by the Wasserstein gradient flow equation. Notably, Ma & Ying (2022, Theorem 5) shows the convergence (without explicit rate) of the flow $(\nu_t, \mu_t)_{t \geq 0}$ to the unique MNE under these separated dynamics.

On the other hand, Lu (2023) examines the situation where the players' flows evolve at varying speeds but with finite timescale separation, meaning that neither player has reached equilibrium. Here, Lu (2023, Theorem 2.1) proves exponential convergence of the finitely timescale-separated Wasserstein gradient flow to the unique MNE, with the convergence rate depending upon regularization and timescale separation parameters. In contrast to Ma & Ying (2022); Lu (2023), we prove that the Fisher-Rao gradient flow (5) converges exponentially fast to the unique MNE while players' dynamics converge at the same speed.

Other works such as (Domingo-Enrich et al., 2020; Wang & Chizat, 2023) focused on the convergence of the Wasserstein-Fisher-Rao (WFR) gradient flow, combining both the Wasserstein gradient flow (allowing particles to diffuse in space) and the Fisher-Rao gradient flow (forcing particles to evade low probability regions). Assuming that $F$ is bilinear and $\sigma = 0$, Domingo-Enrich et al. (2020) investigates the Wasserstein-Fisher-Rao gradient flow's convergence (without giving explicit rates). For $t_0 > 0$ (dependent on the parameters governing the individual contributions of the Wasserstein and Fisher-Rao components in the WFR gradient flow), and under circumstances where the Fisher-Rao component predominates over the Wasserstein component, Domingo-Enrich et al. (2020) prove a characterization result of $\epsilon$-approximate MNE for the WFR gradient flow. More precisely, Domingo-Enrich et al. (2020, Theorem 2) establishes that the pair $\left( \frac{1}{t_0} \int_0^{t_0} \nu_s \mathrm{d}s, \frac{1}{t_0} \int_0^{t_0} \mu_s \mathrm{d}s \right)$ is an $\epsilon$-approximate MNE of the game, i.e., $\mathrm{NI} \left( \frac{1}{t_0} \int_0^{t_0} \nu_s \mathrm{d}s, \frac{1}{t_0} \int_0^{t_0} \mu_s \mathrm{d}s \right) \leq \epsilon$, for any arbitrarily chosen $\epsilon > 0$.

Lastly, Wang & Chizat (2023) introduces a proximal point method that can be viewed as a discrete-time counterpart of the WFR gradient flow. Working within the framework of bilinear $F$, with $\sigma = 0$, and unique MNE, Wang & Chizat (2023, Theorem 2.2) establishes the local exponential convergence of the iterates to the unique MNE of the game with respect to the NI error and the WFR distance, provided that the initialization is done in close vicinity to the MNE. The algorithm proposed by Wang & Chizat (2023) assumes that both players update the positions and weights of the particles simultaneously. However, this is not the only possible discretization for the WFR gradient flow. As highlighted in Lascu et al. (2024) for the discrete-time Fisher-Rao gradient flow, the convergence rates of the dynamics in a game differ depending on the order the players move: either simultaneously (players move at the same time) or sequentially (each player moves upon observing the opponents' moves). Although the continuous-time analysis of the gradient flow does not capture these two situations, it can serve as a guide for designing discrete-time approximations of implementable algorithms.

Moreover, it is argued in Wang & Chizat (2023) that the results of Theorem 2.2 hold only for discrete-time algorithms, and do not imply the convergence of the continuous-time WFR gradient flow because sending the step-sizes of the time discretization to zero will force the initialization $(\nu_0, \mu_0)$ to be already an MNE (see the discussion after Theorem 2.2). Unlike the assumption in Wang & Chizat (2023) that guarantees convergence of the discrete-time algorithm as long as the initialization is close to the MNE, i.e., $\mathrm{NI}(\nu_0, \mu_0) \leq r_0$, for some $r_0 > 0$, depending on the step-sizes, our warm start condition (Assumption 4) for the Fisher-Rao gradient flow only imposes absolute continuity and comparability with *a priori* known reference measures $\pi$ and $\rho$.

Furthermore, Wang & Chizat (2023) discuss that if $(\nu_0, \mu_0)$ is supported on the entire space $\mathcal{X}$ and $\mathcal{Y}$, respectively, then it is expected that the convergence of the discrete-time Fisher-Rao dynamics to an MNE occurs at sub-linear rate in the worst case, and not an exponential rate (see Appendix D). Also, if the strategy spaces $\mathcal{X}$ and $\mathcal{Y}$ are finite and the initialization $(\nu_0, \mu_0)$ is supported on a large number of points uniformly covering $\mathcal{X}$ and $\mathcal{Y}$, respectively, there is no guarantee for last-iterate convergence of discrete-time Fisher-Rao dynamics to an MNE of $V^0$. As showed by Wei et al. (2021); Daskalakis & Panageas (2019), last-iterate convergence guarantees for games on finite strategy spaces $\mathcal{X}$ and $\mathcal{Y}$ hold under the assumption that the MNE is unique, which may not hold for $V^0$.

While it seems natural to combine the Wasserstein and the Fisher-Rao gradient flows, rigorously proving explicit convergence rates for the continuous-time WFR gradient flow for games such as (1) is a challenging and, to our knowledge, still open problem. In this work, we provide an additional step by establishing last-iterate exponential convergence of the Fisher-Rao gradient flow to the unique MNE measured in both KL divergence and NI error (see Theorem 2.3).

## 2 Main results

As we explained in the introduction, we study the convergence of the FR gradient flow to the unique MNE of the entropy-regularized two-player zero-sum game given by (1), where $F : \mathcal{P}(\mathcal{X}) \times \mathcal{P}(\mathcal{Y}) \to \mathbb{R}$ is a non-linear function and $\sigma > 0$. Throughout the paper, we have the following assumptions.

**Assumption 1** (Convexity-concavity of $F$). *Suppose $F$ admits first order flat derivatives with respect to both $\nu$ and $\mu$ as stated in Definition B.1. Furthermore, suppose that $F$ is convex in $\nu$ and concave in $\mu$, i.e., for any $\nu, \nu' \in \mathcal{P}(\mathcal{X})$ and any $\mu, \mu' \in \mathcal{P}(\mathcal{Y})$, we have*

$$F(\nu', \mu) - F(\nu, \mu) \geq \int_{\mathcal{X}} \frac{\delta F}{\delta \nu}(\nu, \mu, x)(\nu' - \nu)(\mathrm{d}x), \tag{10}$$

$$F(\nu, \mu') - F(\nu, \mu) \leq \int_{\mathcal{Y}} \frac{\delta F}{\delta \mu}(\nu, \mu, y)(\mu' - \mu)(\mathrm{d}y). \tag{11}$$

**Assumption 2** (Boundedness of first order flat derivatives). *Suppose $F$ admits first order flat derivatives with respect to both $\nu$ and $\mu$ as stated in Definition B.1 and there exist constants $C_\nu, C_\mu > 0$ such that for all $(\nu, \mu) \in \mathcal{P}(\mathcal{X}) \times \mathcal{P}(\mathcal{Y})$ and for all $(x, y) \in \mathcal{X} \times \mathcal{Y}$, we have*

$$\left| \frac{\delta F}{\delta \nu}(\nu, \mu, x) \right| \leq C_\nu, \quad \left| \frac{\delta F}{\delta \mu}(\nu, \mu, y) \right| \leq C_\mu.$$

**Assumption 3** (Boundedness of second order flat derivatives). *Suppose $F$ admits second order flat derivatives as stated in Definition B.1 and there exist constants $C_{\nu,\nu}, C_{\mu,\mu}, C_{\nu,\mu}, C_{\mu,\nu} > 0$ such that for all $(\nu, \mu) \in \mathcal{P}(\mathcal{X}) \times \mathcal{P}(\mathcal{Y})$ and for all $(x, y), (x', y') \in \mathcal{X} \times \mathcal{Y}$, we have*

$$\left| \frac{\delta^2 F}{\delta \nu^2}(\nu, \mu, x, x') \right| \leq C_{\nu,\nu}, \quad \left| \frac{\delta^2 F}{\delta \mu^2}(\nu, \mu, y, y') \right| \leq C_{\mu,\mu},$$

$$\left| \frac{\delta^2 F}{\delta \nu \delta \mu}(\nu, \mu, y, x) \right| \leq C_{\nu,\mu}, \quad \left| \frac{\delta^2 F}{\delta \mu \delta \nu}(\nu, \mu, x, y) \right| \leq C_{\mu,\nu}.$$

Note that the order of the flat derivatives in $\nu$ and $\mu$ can be interchanged due to Lascu et al. (2023, Lemma B.2). Using Assumption 3, it is straightforward to check that there exist constants $C'_\nu, C'_\mu > 0$ such that for all $(\nu, \mu) \in \mathcal{P}_{\mathrm{ac}}(\mathcal{X}) \times \mathcal{P}_{\mathrm{ac}}(\mathcal{Y})$, $(\nu', \mu') \in \mathcal{P}_{\mathrm{ac}}(\mathcal{X}) \times \mathcal{P}_{\mathrm{ac}}(\mathcal{Y})$ and all $(x, y) \in \mathcal{X} \times \mathcal{Y}$, we have that

$$\left| \frac{\delta F}{\delta \nu}(\nu, \mu, x) - \frac{\delta F}{\delta \nu}(\nu', \mu', x) \right| \leq C'_\nu \left( \mathrm{TV}(\nu, \nu') + \mathrm{TV}(\mu, \mu') \right), \tag{12}$$

$$\left| \frac{\delta F}{\delta \mu}(\nu, \mu, y) - \frac{\delta F}{\delta \mu}(\nu', \mu', y) \right| \leq C'_\mu \left( \mathrm{TV}(\nu, \nu') + \mathrm{TV}(\mu, \mu') \right). \tag{13}$$

**Remark 2.1.** *An example of a function $F$ which satisfies Assumptions 1, 2, 3 is $F(\nu, \mu) = \int_{\mathcal{Y}} \int_{\mathcal{X}} f(x, y) \nu(\mathrm{d}x) \mu(\mathrm{d}y)$, for a function $f : \mathcal{X} \times \mathcal{Y} \to \mathbb{R}$ which is bounded but possibly non-convex-non-concave. Indeed, Assumption 1 is trivially satisfied by such $F$, while Assumptions 2 and 3 hold due to the boundedness of $f$. This type of objective function $F$ is prototypical in applications including the training of GANs (see, e.g., Arjovsky et al. (2017); Hsieh et al. (2019)) and distributionally robust optimization (see, e.g, Madry et al. (2018); Sinha et al. (2018)).*

*Another example that can be viewed as a particular case of our general framework is the objective function of Wasserstein-GANs (WGANs) with gradient penalty (Gulrajani et al., 2017; Petzka et al., 2018). Following Conforti et al. (2023); Kazeykina et al. (2022), GANs can be framed as a min-max problem over the space of probability measures. Consider the following class of parameterized functions as the class of choices for the discriminator:*

$$\{x \mapsto \mathbb{E}[f(Y, x)] : \mathrm{Law}(Y) = \mu \in \mathcal{P}(\mathcal{Y})\}.$$

*The map $x \mapsto \mathbb{E}_{Y \sim \mu}[f(Y, x)]$ is the output of a two-layer neural network with parameters $Y$ and bounded, continuous and non-constant activation function $f$. The objective function of the GAN reads*

$$\min_{\nu \in \mathcal{P}(\mathcal{X})} \max_{\mu \in \mathcal{P}(\mathcal{Y})} \int_{\mathcal{X}} \mathbb{E}_{Y \sim \mu}[f(Y, x)](\nu - \bar{\nu})(\mathrm{d}x) = \min_{\nu \in \mathcal{P}(\mathcal{X})} \max_{\mu \in \mathcal{P}(\mathcal{Y})} \int_{\mathcal{Y}} \int_{\mathcal{X}} f(x, y)(\nu - \bar{\nu})(\mathrm{d}x) \mu(\mathrm{d}y),$$

*where $\bar{\nu}$ is the distribution of the true data. Assuming that $\mathcal{X} \ni x \mapsto f(x, y) \in \mathbb{R}$ is differentiable and 1-Lipschitz, we retrieve the WGAN. The gradient penalty regularization terms proposed by Gulrajani et al. (2017); Petzka et al. (2018) are added in order to ensure that the discriminator remains 1-Lipschitz along the interpolation between the generated and true data. Thus, the objective function can be expressed as*

$$\min_{\nu \in \mathcal{P}(\mathcal{X})} \max_{\mu \in \mathcal{P}(\mathcal{Y})} \left\{ \int_{\mathcal{Y}} \int_{\mathcal{X}} f(x, y)(\nu - \bar{\nu})(\mathrm{d}x) \mu(\mathrm{d}y) + \lambda t \int_{\mathcal{Y}} \int_{\mathcal{X}} \left( \|\nabla_x f(x, y)\| - 1 \right)^2 (\nu - \bar{\nu})(\mathrm{d}x) \mu(\mathrm{d}y) \right.$$

$$\left. + \lambda(1 - t) g \left( \int_{\mathcal{Y}} \int_{\mathcal{X}} \left( \|\nabla_x f(x, y)\| - 1 \right)^2 \bar{\nu}(\mathrm{d}x) \mu(\mathrm{d}y) \right) \right\},$$

*where $g : \mathbb{R} \to \mathbb{R}$ is a twice differentiable concave function with bounded derivatives, $\lambda > 0$ is a regularization parameter and $t \in [0, 1]$. Since $g$ is concave, the term inside $g$ is linear in $\mu$ and independent of $\nu$, it follows that the third term in the optimization problem is concave in $\mu$. This fact together with the linearity in $\nu$ and $\mu$ of the first two terms shows that the objective function satisfies Assumption 1. Since $f$ is bounded, differentiable and 1-Lipschitz, and $g$ is twice differentiable with bounded derivatives, Assumptions 2 and 3 are also satisfied.*

**Assumption 4** (Ratio condition). *Suppose $(\nu_0, \mu_0) \in \mathcal{P}(\mathcal{X}) \times \mathcal{P}(\mathcal{Y})$ are absolutely continuous and comparable with $\pi$ and $\rho$, respectively, in the sense that*

1. *There exist constants $r_\nu, r_\mu > 0$ such that*

$$\inf_{x \in \mathcal{X}} \frac{\nu_0(x)}{\pi(x)} \geq r_\nu, \quad \inf_{y \in \mathcal{Y}} \frac{\mu_0(y)}{\rho(y)} \geq r_\mu. \tag{14}$$

2. *There exist constants $R_\nu, R_\mu > 1$ such that*

$$\sup_{x \in \mathcal{X}} \frac{\nu_0(x)}{\pi(x)} \leq R_\nu, \quad \sup_{y \in \mathcal{Y}} \frac{\mu_0(y)}{\rho(y)} \leq R_\mu. \tag{15}$$

It can be proved (see, e.g., Lascu et al. (2023, Proposition A.1)) that the MNE $(\nu_\sigma^*, \mu_\sigma^*)$ of (1) satisfies the fixed point equations

$$\nu_\sigma^*(x) = \frac{1}{Z(\nu_\sigma^*, \mu_\sigma^*)} \exp \left( -\frac{2}{\sigma^2} \frac{\delta F}{\delta \nu}(\nu_\sigma^*, \mu_\sigma^*, x) - U^\pi(x) \right), \tag{16}$$

$$\mu_\sigma^*(y) = \frac{1}{Z'(\nu_\sigma^*, \mu_\sigma^*)} \exp \left( \frac{2}{\sigma^2} \frac{\delta F}{\delta \mu}(\nu_\sigma^*, \mu_\sigma^*, y) - U^\rho(y) \right), \tag{17}$$

where $Z(\nu_\sigma^*, \mu_\sigma^*)$ and $Z'(\nu_\sigma^*, \mu_\sigma^*)$ are normalizing constants.

Combining (16) and (17) and Assumption 2, we deduce that Assumption 4 is equivalent to assuming that there exist constants $\bar{r}_\nu, \bar{r}_\mu > 0$ and $\bar{R}_\nu, \bar{R}_\mu > 1$ such that for all $(x, y) \in \mathcal{X} \times \mathcal{Y}$,

$$\bar{r}_\nu \leq \frac{\nu_0(x)}{\nu_\sigma^*(x)} \leq \bar{R}_\nu, \quad \bar{r}_\mu \leq \frac{\mu_0(y)}{\mu_\sigma^*(y)} \leq \bar{R}_\mu. \tag{18}$$

We emphasize that Assumption 4 is natural in the context of Fisher-Rao flows. From (5), we observe that the support of the measures along the gradient flow does not increase. Thus, it is essential that the initialization is comparable with the MNE (cf. (18)) as reflected in Assumption 4. It is observed in Liu et al. (2023), that Assumption 4 can be understood as a "warm start" type of condition.

Returning to the question of flat differentiability of $V^\sigma$, which was raised in Subsection 1.2, if we assume that $F$ is flat differentiable with respect to both $\nu$ and $\mu$ (see Assumption 1), then the maps $(\nu, \mu, x) \mapsto a(\nu, \mu, x) :=$ $\frac{\delta F}{\delta \nu}(\nu, \mu, x) + \frac{\sigma^2}{2} \log\left(\frac{\nu(x)}{\pi(x)}\right) - \frac{\sigma^2}{2} D_{KL}(\nu|\pi)$ and $(\nu, \mu, y) \mapsto b(\nu, \mu, y) := \frac{\delta F}{\delta \mu}(\nu, \mu, y) - \frac{\sigma^2}{2} \log\left(\frac{\mu(y)}{\rho(y)}\right) + \frac{\sigma^2}{2} D_{KL}(\mu|\rho)$ are well-defined and formally correspond to the flat derivatives $\frac{\delta V^\sigma}{\delta \nu}(\nu, \mu, \cdot)$ and $\frac{\delta V^\sigma}{\delta \mu}(\nu, \mu, \cdot)$, respectively, for those measures $\nu$ and $\mu$ for which such derivatives exist (note that we will only need to consider $a$ and $b$ along our gradient flow $(\nu_t, \mu_t)_{t \geq 0}$, so our argument can be interpreted as stating that, while $V^\sigma$ is not flat differentiable everywhere, it is indeed flat differentiable along our gradient flow). The relative entropy terms $D_{KL}$ appear in the definition of $a$ and $b$ as normalizing constants to ensure that $\int_{\mathcal{X}} a(\nu, \mu, x)\nu(dx) = 0$ and $\int_{\mathcal{Y}} b(\nu, \mu, y)\mu(dy) = 0$, since we adopt the convention that the flat derivatives of $F$ are uniquely defined up to an additive shift (see Definition B.1 in Section B). Motivated by this discussion, we define the Fisher-Rao gradient flow $(\nu_t, \mu_t)_{t \geq 0}$ on the space $(\mathcal{P}_{ac}(\mathcal{X}) \times \mathcal{P}_{ac}(\mathcal{Y}), FR)$ by

$$\begin{cases} \partial_t \nu_t(x) = -a(\nu_t, \mu_t, x)\nu_t(x), \\ \partial_t \mu_t(y) = b(\nu_t, \mu_t, y)\mu_t(y), \end{cases} \tag{19}$$

with initial condition $(\nu_0, \mu_0) \in \mathcal{P}_{ac}(\mathcal{X}) \times \mathcal{P}_{ac}(\mathcal{Y})$. We will establish the existence of a solution to (19) in Theorem 2.2. As we will demonstrate, the flow of densities $(\nu_t, \mu_t)_{t \geq 0}$ is differentiable in time (cf. equation (35)), and hence the solution to (19) can be interpreted just as a classical solution to an ordinary differential equation.

The following result extends Liu et al. (2023, Theorem 2.1) to the case of two-player zero-sum games by showing that the Fisher-Rao gradient flow (19) admits a unique solution $(\nu_t, \mu_t)_{t \geq 0}$.

**Theorem 2.2.** *Suppose that Assumption 2, 3 and condition* (15) *from Assumption 4 hold. Then for each $(\nu_0, \mu_0) \in \mathcal{P}_{ac}(\mathcal{X}) \times \mathcal{P}_{ac}(\mathcal{Y})$, there exists a unique pair of continuous and differentiable in time flows $(\nu_t, \mu_t)_{t \in [0, \infty)} \in \mathcal{P}_{ac}(\mathcal{X}) \times \mathcal{P}_{ac}(\mathcal{Y})$ satisfying the system of equations* (19). *Moreover, for $t > 0$,*

$$D_{KL}(\nu_t|\pi) \leq 2\log R_\nu + \frac{4}{\sigma^2}C_\nu, \quad D_{KL}(\mu_t|\rho) \leq 2\log R_\mu + \frac{4}{\sigma^2}C_\mu \tag{20}$$

*and there exist constants $R_{1,\nu}, R_{1,\mu} > 1$ such that for all $t > 0$,*

$$\sup_{x \in \mathcal{X}} \frac{\nu_t(x)}{\pi(x)} \leq R_{1,\nu}, \quad \sup_{y \in \mathcal{Y}} \frac{\mu_t(y)}{\rho(y)} \leq R_{1,\mu}. \tag{21}$$

*Additionally, if condition* (14) *from Assumption 4 holds, then there exist constants $r_{1,\nu}, r_{1,\mu} > 0$ such that for all $t > 0$,*

$$\inf_{x \in \mathcal{X}} \frac{\nu_t(x)}{\pi(x)} \geq r_{1,\nu}, \quad \inf_{y \in \mathcal{Y}} \frac{\mu_t(y)}{\rho(y)} \geq r_{1,\mu}. \tag{22}$$

The bounds obtained in this theorem allow us to prove that the map $t \mapsto D_{KL}(\nu_\sigma^*|\nu_t) + D_{KL}(\mu_\sigma^*|\mu_t)$ is differentiable along the FR dynamics (19).

Next, we state the other main result of this paper. We prove two different types of convergence results for the FR dynamics (19) in min-max games given by (1), one in terms of the players' strategies and the other

in terms of the payoff function. Whereas the proof of the existence of the flow (Theorem 2.2) follows a route similar to Liu et al. (2023), the proof of convergence in Theorem 2.3 is significantly different from Liu et al. (2023) since, as indicated in the introduction, in the present paper, convergence has to be studied with respect to appropriately chosen Lyapunov functions and, moreover, we do not rely on the Polyak-Łojasiewicz inequality.

**Theorem 2.3.** *Suppose that Assumption 2, 3 and 4 hold and let $(\nu, \mu) \in \mathcal{P}_{ac}(\mathcal{X}) \times \mathcal{P}_{ac}(\mathcal{Y})$. Then the map $t \mapsto \mathrm{D}_{\mathrm{KL}}(\nu|\nu_t) + \mathrm{D}_{\mathrm{KL}}(\mu|\mu_t)$ is differentiable along the FR dynamics (19). Suppose furthermore that Assumption 1 holds. Then, for all $t > 0$, we have*

$$\mathrm{D}_{\mathrm{KL}}(\nu_\sigma^*|\nu_t) + \mathrm{D}_{\mathrm{KL}}(\mu_\sigma^*|\mu_t) \leq e^{-\frac{\sigma^2}{2}t} \left(\mathrm{D}_{\mathrm{KL}}(\nu_\sigma^*|\nu_0) + \mathrm{D}_{\mathrm{KL}}(\mu_\sigma^*|\mu_0)\right),$$

$$\mathrm{NI}(\nu_t, \mu_t) \leq 2C_\sigma e^{-\frac{\sigma^2}{4}t} \sqrt{\mathrm{D}_{\mathrm{KL}}(\nu_\sigma^*|\nu_0) + \mathrm{D}_{\mathrm{KL}}(\mu_\sigma^*|\mu_0)}. \tag{23}$$

In game theoretic language, the first result of Theorem 2.3 says that convergence of the FR dynamics (19) to the unique MNE $(\nu_\sigma^*, \mu_\sigma^*)$ in terms of the strategies $\nu_t$ and $\mu_t$ is achieved with exponential rate depending on $\sigma$, while the second result shows exponential convergence in terms of the payoff function $V^\sigma$.

**Remark 2.4.** *Exponential convergence of the single mean-field birth-death flow $(\nu_t)_{t \geq 0}$ with respect to $\mathrm{D}_{\mathrm{KL}}(\nu_\sigma^*|\nu_t)$ can be shown to also hold in the setting of Liu et al. (2023), however it was not studied in Liu et al. (2023), which considered only convergence of $V^\sigma(\nu_t)$ for a convex energy function $V^\sigma : \mathcal{P}(\mathbb{R}^d) \to \mathbb{R}$.*

## 3 Proof of Theorem 2.3

Before we present the proof of Theorem 2.3, we begin with a technical lemma, which extends Liu et al. (2023, Lemma 2.5) to the min-max games setup, and which is proved in Section A. The proof of Theorem 2.3 is done in two steps:

- First, we show that, for fixed $(\nu, \mu)$, the map $t \mapsto \mathrm{D}_{\mathrm{KL}}(\nu|\nu_t) + \mathrm{D}_{\mathrm{KL}}(\mu|\mu_t)$ is differentiable when $(\nu_t, \mu_t)_{t \geq 0}$ is defined as the FR flow (19).

- Second, we show that $\frac{\mathrm{d}}{\mathrm{d}t} \left(\mathrm{D}_{\mathrm{KL}}(\nu_\sigma^*|\nu_t) + \mathrm{D}_{\mathrm{KL}}(\mu_\sigma^*|\mu_t)\right)$ is bounded from above by $-\frac{\sigma^2}{2} \left(\mathrm{D}_{\mathrm{KL}}(\nu_\sigma^*|\nu_t) + \mathrm{D}_{\mathrm{KL}}(\mu_\sigma^*|\mu_t)\right)$, and then we apply Gronwall's inequality to obtain exponential convergence. Subsequently, we establish exponential convergence for $t \mapsto \mathrm{NI}(\nu_t, \mu_t)$.

**Lemma 3.1** (Relative $\sigma$-strong-convexity-concavity to $\mathrm{D}_{\mathrm{KL}}$). *For $V^\sigma$ given by (1), if Assumption 1 holds, then $V^\sigma$ satisfies the following inequalities for all $(\nu, \mu), (\nu', \mu') \in \mathcal{P}_{ac}(\mathcal{X}) \times \mathcal{P}_{ac}(\mathcal{Y})$:*

$$V^\sigma(\nu', \mu) - V^\sigma(\nu, \mu) \geq \int_{\mathcal{X}} a(\nu, \mu, x)(\nu' - \nu)(\mathrm{d}x) + \frac{\sigma^2}{2} \mathrm{D}_{\mathrm{KL}}(\nu'|\nu),$$

$$V^\sigma(\nu, \mu') - V^\sigma(\nu, \mu) \leq \int_{\mathcal{Y}} b(\nu, \mu, y)(\mu' - \mu)(\mathrm{d}y) - \frac{\sigma^2}{2} \mathrm{D}_{\mathrm{KL}}(\mu'|\mu).$$

*Proof of Theorem 2.3. Step 1: Differentiability of $\mathrm{D}_{\mathrm{KL}}$ with respect to the FR flow (19):* Suppose that Assumption 2, 3 and 4 hold. In order to show the differentiability of $t \mapsto \mathrm{D}_{\mathrm{KL}}(\nu|\nu_t)$ for fixed $\nu \in \mathcal{P}_{ac}(\mathcal{X})$ with respect to (19), it suffices to show that there exists an integrable function $f : \mathcal{X} \to \mathbb{R}$ such that

$$\left| \partial_t \left( \nu(x) \log \frac{\nu(x)}{\nu_t(x)} \right) \right| \leq f(x),$$

for all $t \geq 0$. Indeed, using (19) and (36), we have that

$$\left| \partial_t \left( \nu(x) \log \frac{\nu(x)}{\nu_t(x)} \right) \right| = \left| \nu(x) \frac{\partial_t \nu_t(x)}{\nu_t(x)} \right| = \left| \nu(x) \left( \frac{\delta F}{\delta \nu}(\nu_t, \mu_t, x) + \frac{\sigma^2}{2} \log \left( \frac{\nu_t(x)}{\pi(x)} \right) - \frac{\sigma^2}{2} \mathrm{D}_{\mathrm{KL}}(\nu_t|\pi) \right) \right|$$

$$\leq \left( 3C_\nu + \frac{\sigma^2}{2} \left( \max\{|\log r_{1,\nu}|, \log R_{1,\nu}\} + 2 \log R_\nu \right) \right) \nu(x) := f(x).$$

An identical argument gives the differentiability of $t \mapsto \mathrm{D_{KL}}(\mu|\mu_t)$ for fixed $\mu \in \mathcal{P}_{\mathrm{ac}}(\mathcal{Y})$. Then, we have that the map $t \mapsto \mathrm{D_{KL}}(\nu|\nu_t) + \mathrm{D_{KL}}(\mu|\mu_t)$ is differentiable.

*Step 2: Convergence of the FR flow:* Since $t \mapsto \mathrm{D_{KL}}(\nu|\nu_t) + \mathrm{D_{KL}}(\mu|\mu_t)$ is differentiable, we have that

$$
\begin{aligned}
\frac{\mathrm{d}}{\mathrm{d}t} \left( \mathrm{D_{KL}}(\nu|\nu_t) + \mathrm{D_{KL}}(\mu|\mu_t) \right) &= \int_{\mathcal{X}} \partial_t \left( \nu(x) \log \frac{\nu(x)}{\nu_t(x)} \right) \mathrm{d}x + \int_{\mathcal{Y}} \partial_t \left( \mu(y) \log \frac{\mu(y)}{\mu_t(y)} \right) \mathrm{d}y \\
&= - \int_{\mathcal{X}} (\nu(x) - \nu_t(x)) \frac{\partial_t \nu_t(x)}{\nu_t(x)} \mathrm{d}x - \int_{\mathcal{Y}} (\mu(y) - \mu_t(y)) \frac{\partial_t \mu_t(y)}{\mu_t(y)} \mathrm{d}y \\
&= \int_{\mathcal{X}} a(\nu_t, \mu_t, x)(\nu - \nu_t)(\mathrm{d}x) - \int_{\mathcal{Y}} b(\nu_t, \mu_t, y)(\mu - \mu_t)(\mathrm{d}y),
\end{aligned}
$$

where in the second equality we used the fact that $\int_{\mathcal{X}} \partial_t \nu_t(x) \mathrm{d}x = \int_{\mathcal{Y}} \partial_t \mu_t(y) \mathrm{d}y = 0$.

If $\sigma > 0$ and Assumption 1 holds then, using Lemma 3.1 in the equation above, we obtain

$$
\frac{\mathrm{d}}{\mathrm{d}t} \left( \mathrm{D_{KL}}(\nu|\nu_t) + \mathrm{D_{KL}}(\mu|\mu_t) \right) \leq V^\sigma(\nu, \mu_t) - V^\sigma(\nu_t, \mu) - \frac{\sigma^2}{2} \mathrm{D_{KL}}(\nu|\nu_t) - \frac{\sigma^2}{2} \mathrm{D_{KL}}(\mu|\mu_t). \tag{24}
$$

Setting $(\nu, \mu) = (\nu_\sigma^*, \mu_\sigma^*)$ in (24) and using the saddle point condition (2) gives

$$
\frac{\mathrm{d}}{\mathrm{d}t} \left( \mathrm{D_{KL}}(\nu_\sigma^*|\nu_t) + \mathrm{D_{KL}}(\mu_\sigma^*|\mu_t) \right) \leq -\frac{\sigma^2}{2} \mathrm{D_{KL}}(\nu_\sigma^*|\nu_t) - \frac{\sigma^2}{2} \mathrm{D_{KL}}(\mu_\sigma^*|\mu_t).
$$

Hence, by Gronwall's inequality, we obtain that

$$
\mathrm{D_{KL}}(\nu_\sigma^*|\nu_t) + \mathrm{D_{KL}}(\mu_\sigma^*|\mu_t) \leq e^{-\frac{\sigma^2}{2}t} \left( \mathrm{D_{KL}}(\nu_\sigma^*|\nu_0) + \mathrm{D_{KL}}(\mu_\sigma^*|\mu_0) \right). \tag{25}
$$

To prove convergence in terms of the NI error, first observe that, for any $(\nu, \mu) \in \mathcal{P}(\mathcal{X}) \times \mathcal{P}(\mathcal{Y})$, we can write

$$
V^\sigma(\nu_t, \mu) - V^\sigma(\nu, \mu_t) = V^\sigma(\nu_t, \mu) - V^\sigma(\nu_\sigma^*, \mu) + V^\sigma(\nu_\sigma^*, \mu) - V^\sigma(\nu, \mu_\sigma^*) + V^\sigma(\nu, \mu_\sigma^*) - V^\sigma(\nu, \mu_t).
$$

By Theorem 2.2 and Assumption 2, there exist $C_{1,\sigma}, C_{2,\sigma} > 0$ depending on $\sigma$ such that

$$
\left| \frac{\delta V^\sigma}{\delta \nu}(\nu_t, \mu, x) \right| \leq C_{1,\sigma}, \quad \left| \frac{\delta V^\sigma}{\delta \mu}(\nu, \mu_t, y) \right| \leq C_{2,\sigma},
$$

for all $(\nu, \mu) \in \mathcal{P}(\mathcal{X}) \times \mathcal{P}(\mathcal{Y})$, and all $(x, y) \in \mathcal{X} \times \mathcal{Y}$. Then, by Lemma 3.1, we have

$$
\begin{aligned}
V^\sigma(\nu_t, \mu) - V^\sigma(\nu_\sigma^*, \mu) &\leq \int_{\mathcal{X}} \frac{\delta V^\sigma}{\delta \nu}(\nu_t, \mu, x)(\nu_t - \nu_\sigma^*)(\mathrm{d}x) - \frac{\sigma^2}{2} \mathrm{D_{KL}}(\nu_\sigma^*|\nu_t) \\
&\leq C_{1,\sigma} \mathrm{TV}(\nu_\sigma^*, \nu_t) \leq C_{1,\sigma} \sqrt{2 \mathrm{D_{KL}}(\nu_\sigma^*|\nu_t)},
\end{aligned}
$$

and

$$
\begin{aligned}
V^\sigma(\nu, \mu_\sigma^*) - V^\sigma(\nu, \mu_t) &\leq \int_{\mathcal{Y}} \frac{\delta V^\sigma}{\delta \mu}(\nu, \mu_t, y)(\mu_\sigma^* - \mu_t)(\mathrm{d}y) - \frac{\sigma^2}{2} \mathrm{D_{KL}}(\mu_\sigma^*|\mu_t) \\
&\leq C_{2,\sigma} \mathrm{TV}(\mu_\sigma^*, \mu_t) \leq C_{2,\sigma} \sqrt{2 \mathrm{D_{KL}}(\mu_\sigma^*|\mu_t)},
\end{aligned}
$$

where the last inequalities hold due to Pinsker's inequality and since $\frac{\sigma^2}{2} \mathrm{D_{KL}}(\nu_\sigma^*|\nu_t), \frac{\sigma^2}{2} \mathrm{D_{KL}}(\mu_\sigma^*|\mu_t) \geq 0$, for all $t \geq 0$. Setting $C_\sigma := \max\{C_{1,\sigma}, C_{2,\sigma}\}$, and using the inequality $\sqrt{a} + \sqrt{b} \leq \sqrt{2(a+b)}$, we get

$$
\begin{aligned}
V^\sigma(\nu_t, \mu) - V^\sigma(\nu, \mu_t) &\leq 2C_\sigma \sqrt{\mathrm{D_{KL}}(\nu_\sigma^*|\nu_t) + \mathrm{D_{KL}}(\mu_\sigma^*|\mu_t)} + V^\sigma(\nu_\sigma^*, \mu) - V^\sigma(\nu, \mu_\sigma^*) \tag{26} \\
&\leq 2C_\sigma e^{-\frac{\sigma^2}{4}t} \sqrt{\mathrm{D_{KL}}(\nu_\sigma^*|\nu_0) + \mathrm{D_{KL}}(\mu_\sigma^*|\mu_0)} + V^\sigma(\nu_\sigma^*, \mu) - V^\sigma(\nu, \mu_\sigma^*),
\end{aligned}
$$

where the last inequality follows from (25). Note that since $(\nu_\sigma^*, \mu_\sigma^*)$ is the MNE of (1), we have

$$
\max_\mu V^\sigma(\nu_\sigma^*, \mu) = \min_\nu V^\sigma(\nu, \mu_\sigma^*) = V^\sigma(\nu_\sigma^*, \mu_\sigma^*),
$$

Therefore, maximizing over $(\nu, \mu)$ in (26) gives

$$\mathrm{NI}(\nu_t, \mu_t) = \max_{\mu} V^{\sigma}(\nu_t, \mu) - \min_{\nu} V^{\sigma}(\nu, \mu_t) \leq 2C_{\sigma} e^{-\frac{\sigma^2}{4} t} \sqrt{\mathrm{D_{KL}}(\nu_{\sigma}^* | \nu_0) + \mathrm{D_{KL}}(\mu_{\sigma}^* | \mu_0)}.$$

Finally, observe that, by (18), we have

$$\mathrm{D_{KL}}(\nu_{\sigma}^* | \nu_0) + \mathrm{D_{KL}}(\mu_{\sigma}^* | \mu_0) < \infty.$$

$\square$

## Acknowledgements

R-AL was supported by the EPSRC Centre for Doctoral Training in Mathematical Modelling, Analysis and Computation (MAC-MIGS) funded by the UK Engineering and Physical Sciences Research Council (grant EP/S023291/1), Heriot-Watt University and the University of Edinburgh. LS acknowledges the support of the UKRI Prosperity Partnership Scheme (FAIR) under EPSRC Grant EP/V056883/1 and the Alan Turing Institute.

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

# A  Technical results and proofs

In this section, we present the proofs of the remaining results formulated in Section 2 of the paper.

## A.1  Proof of Lemma 3.1.

In this subsection, we present the proof of Lemma 3.1.

*Proof of Lemma 3.1.* Using (10) from Assumption 1, it follows that

$$
\begin{aligned}
V^\sigma(\nu', \mu) - V^\sigma(\nu, \mu) &\geq \int_{\mathcal{X}} \frac{\delta F}{\delta \nu}(\nu, \mu, x)(\nu' - \nu)(\mathrm{d}x) + \frac{\sigma^2}{2} \mathrm{D_{KL}}(\nu'|\pi) - \frac{\sigma^2}{2} \mathrm{D_{KL}}(\nu|\pi) \\
&= \int_{\mathcal{X}} \left( \frac{\delta F}{\delta \nu}(\nu, \mu, x) + \frac{\sigma^2}{2} \log\left( \frac{\nu(x)}{\pi(x)} \right) \right)(\nu' - \nu)(\mathrm{d}x) - \frac{\sigma^2}{2} \int_{\mathcal{X}} \log\left( \frac{\nu(x)}{\pi(x)} \right)(\nu' - \nu)(\mathrm{d}x) \\
&\quad + \frac{\sigma^2}{2} \int_{\mathcal{X}} \log\left( \frac{\nu'(x)}{\pi(x)} \right) \nu'(\mathrm{d}x) - \frac{\sigma^2}{2} \int_{\mathcal{X}} \log\left( \frac{\nu(x)}{\pi(x)} \right) \nu(\mathrm{d}x) \\
&= \int_{\mathcal{X}} \left( \frac{\delta F}{\delta \nu}(\nu, \mu, x) + \frac{\sigma^2}{2} \log\left( \frac{\nu(x)}{\pi(x)} \right) \right)(\nu' - \nu)(\mathrm{d}x) + \frac{\sigma^2}{2} \mathrm{D_{KL}}(\nu'|\nu) \\
&\qquad\qquad\qquad = \int_{\mathcal{X}} a(\nu, \mu, x)(\nu' - \nu)(\mathrm{d}x) + \frac{\sigma^2}{2} \mathrm{D_{KL}}(\nu'|\nu).
\end{aligned}
$$

Similarly, using (11) from Assumption 1, it follows that

$$
\begin{aligned}
V^\sigma(\nu, \mu') - V^\sigma(\nu, \mu) &\leq \int_{\mathcal{Y}} \frac{\delta F}{\delta \mu}(\nu, \mu, y)(\mu' - \mu)(\mathrm{d}y) - \frac{\sigma^2}{2} \mathrm{D_{KL}}(\mu'|\rho) + \frac{\sigma^2}{2} \mathrm{D_{KL}}(\mu|\rho) \\
&= \int_{\mathcal{Y}} \left( \frac{\delta F}{\delta \mu}(\nu, \mu, y) - \frac{\sigma^2}{2} \log\left( \frac{\mu(y)}{\rho(y)} \right) \right)(\mu' - \mu)(\mathrm{d}y) + \frac{\sigma^2}{2} \int_{\mathcal{Y}} \log\left( \frac{\mu(y)}{\rho(y)} \right)(\mu' - \mu)(\mathrm{d}y) \\
&\quad - \frac{\sigma^2}{2} \int_{\mathcal{Y}} \log\left( \frac{\mu'(y)}{\rho(y)} \right) \mu'(\mathrm{d}y) + \frac{\sigma^2}{2} \int_{\mathcal{Y}} \log\left( \frac{\mu(y)}{\rho(y)} \right) \mu(\mathrm{d}y) \\
&= \int_{\mathcal{Y}} \left( \frac{\delta F}{\delta \mu}(\nu, \mu, y) - \frac{\sigma^2}{2} \log\left( \frac{\mu(y)}{\rho(y)} \right) \right)(\mu' - \mu)(\mathrm{d}y) - \frac{\sigma^2}{2} \mathrm{D_{KL}}(\mu'|\mu) \\
&\qquad\qquad\qquad = \int_{\mathcal{Y}} b(\nu, \mu, y)(\mu' - \mu)(\mathrm{d}y) - \frac{\sigma^2}{2} \mathrm{D_{KL}}(\mu'|\mu).
\end{aligned}
$$

$\square$

## A.2 Existence and uniqueness of the FR flow

In this subsection, we present the proof of our main result concerning the existence and uniqueness of the Fisher-Rao (FR) flow, i.e., Theorem 2.2. We construct a Picard iteration which is proved to be well-defined in Lemma A.1. Lemma A.2 shows that the Picard iteration mapping admits a unique fixed point in an appropriate metric. Then in order to conclude the proof of Theorem 2.2 we show the ratio condition (21).

The proof of Theorem 2.2 is an adaptation of the proof of Liu et al. (2023, Theorem 2.1) to the min-max setting (1).

*Proof of Theorem 2.2. Step 1: Existence of the gradient flow and bound* (20) *on* $[0, T]$. In order to prove the existence of a solution $(\nu_t, \mu_t)_{t \geq 0}$ to

$$
\begin{cases}
\partial_t \nu_t(x) = -\left( \frac{\delta F}{\delta \nu}(\nu_t, \mu_t, x) + \frac{\sigma^2}{2} \log\left( \frac{\nu_t(x)}{\pi(x)} \right) - \frac{\sigma^2}{2} \mathrm{D_{KL}}(\nu_t|\pi) \right) \nu_t(x), \\
\partial_t \mu_t(y) = \left( \frac{\delta F}{\delta \mu}(\nu_t, \mu_t, y) - \frac{\sigma^2}{2} \log\left( \frac{\mu_t(y)}{\rho(y)} \right) + \frac{\sigma^2}{2} \mathrm{D_{KL}}(\mu_t|\rho) \right) \mu_t(y),
\end{cases} \tag{27}
$$

we first notice that (27) is equivalent to

$$
\begin{cases}
\partial_t \log \nu_t(x) = -\left( \frac{\delta F}{\delta \nu}(\nu_t, \mu_t, x) + \frac{\sigma^2}{2} \log\left( \frac{\nu_t(x)}{\pi(x)} \right) - \frac{\sigma^2}{2} \mathrm{D_{KL}}(\nu_t|\pi) \right), \\
\partial_t \log \mu_t(y) = \left( \frac{\delta F}{\delta \mu}(\nu_t, \mu_t, y) - \frac{\sigma^2}{2} \log\left( \frac{\mu_t(y)}{\rho(y)} \right) + \frac{\sigma^2}{2} \mathrm{D_{KL}}(\mu_t|\rho) \right).
\end{cases} \tag{28}
$$

By Duhamel's formula, (28) is equivalent to

$$
\begin{cases}
\log \nu_t(x) = e^{-\frac{\sigma^2}{2}t} \log \nu_0(x) - \int_0^t \frac{\sigma^2}{2} e^{-\frac{\sigma^2}{2}(t-s)} \left( \frac{2}{\sigma^2} \frac{\delta F}{\delta \nu}(\nu_s, \mu_s, x) - \log \pi(x) - \mathrm{D_{KL}}(\nu_s|\pi) \right) \mathrm{d}s, \\
\log \mu_t(y) = e^{-\frac{\sigma^2}{2}t} \log \mu_0(x) + \int_0^t \frac{\sigma^2}{2} e^{-\frac{\sigma^2}{2}(t-s)} \left( \frac{2}{\sigma^2} \frac{\delta F}{\delta \nu}(\nu_s, \mu_s, y) + \log \rho(y) + \mathrm{D_{KL}}(\mu_s|\rho) \right) \mathrm{d}s.
\end{cases}
$$

Based on these formulas, we will define a Picard iteration scheme. To this end, let us first fix $T > 0$ and choose a pair of flows of probability measures $(\nu_t^{(0)}, \mu_t^{(0)})_{t\in[0,T]}$ such that

$$\int_0^T \mathrm{D_{KL}}(\nu_s^{(0)}|\pi)\mathrm{d}s < \infty, \quad \int_0^T \mathrm{D_{KL}}(\mu_s^{(0)}|\rho)\mathrm{d}s < \infty.$$

For each $n \geq 1$, we fix $\nu_0^{(n)} = \nu_0^{(0)} = \nu_0$ and $\mu_0^{(n)} = \mu_0^{(0)} = \mu_0$ (with $\nu_0$ and $\mu_0$ satisfying condition (15) from Assumption 4) and define $(\nu_t^{(n)}, \mu_t^{(n)})_{t\in[0,T]}$ by

$$\begin{aligned}
\log \nu_t^{(n)}(x) = e^{-\frac{\sigma^2}{2}t} \log \nu_0(x) \\
- \int_0^t \frac{\sigma^2}{2} e^{-\frac{\sigma^2}{2}(t-s)} \left( \frac{2}{\sigma^2} \frac{\delta F}{\delta \nu}(\nu_s^{(n-1)}, \mu_s^{(n-1)}, x) - \log \pi(x) - \mathrm{D_{KL}}(\nu_s^{(n-1)}|\pi) \right) \mathrm{d}s,
\end{aligned} \quad (29)$$

$$\begin{aligned}
\log \mu_t^{(n)}(y) = e^{-\frac{\sigma^2}{2}t} \log \mu_0(y) \\
+ \int_0^t \frac{\sigma^2}{2} e^{-\frac{\sigma^2}{2}(t-s)} \left( \frac{2}{\sigma^2} \frac{\delta F}{\delta \mu}(\nu_s^{(n-1)}, \mu_s^{(n-1)}, y) + \log \rho(y) + \mathrm{D_{KL}}(\mu_s^{(n-1)}|\rho) \right) \mathrm{d}s.
\end{aligned} \quad (30)$$

We have the following result.

**Lemma A.1.** *The sequence of flows $\left((\nu_t^{(n)}, \mu_t^{(n)})_{t\in[0,T]}\right)_{n=0}^{\infty}$ given by (29) and (30) is well-defined and such that for all $n \geq 1$ and all $t \in [0, T]$ we have*

$$\mathrm{D_{KL}}(\nu_t^{(n)}|\pi) \leq 2 \log R_\nu + \frac{4}{\sigma^2} C_\nu, \quad \mathrm{D_{KL}}(\mu_t^{(n)}|\rho) \leq 2 \log R_\mu + \frac{4}{\sigma^2} C_\mu.$$

*Proof of Lemma A.1.* The proof follows from the same induction argument used to prove Liu et al. (2023, Lemma 3.1). □

For fixed $T > 0$, we consider the sequence of flows $\left((\nu_t^{(n)}, \mu_t^{(n)})_{t\in[0,T]}\right)_{n=0}^{\infty}$ in $\left(\mathcal{P}(\mathcal{X})^{[0,T]} \times \mathcal{P}(\mathcal{Y})^{[0,T]}, \mathcal{TV}^{[0,T]}\right)$, where, for any $(\nu_t, \mu_t)_{t\in[0,T]} \in \mathcal{P}(\mathcal{X})^{[0,T]} \times \mathcal{P}(\mathcal{Y})^{[0,T]}$, the distance $\mathcal{TV}^{[0,T]}$ is defined by

$$\mathcal{TV}^{[0,T]}\left((\nu_t, \mu_t)_{t\in[0,T]}, (\nu_t', \mu_t')_{t\in[0,T]}\right) := \int_0^T \mathrm{TV}(\nu_t, \nu_t')\mathrm{d}t + \int_0^T \mathrm{TV}(\mu_t, \mu_t')\mathrm{d}t.$$

Since $(\mathcal{P}(\mathcal{X}), \mathrm{TV})$ is complete, we can apply the argument from Šiška & Szpruch (2020, Lemma A.5) with $p = 1$ to conclude that $\left(\mathcal{P}(\mathcal{X})^{[0,T]}, \int_0^T \mathrm{TV}(\nu_t, \nu_t')\mathrm{d}t\right)$ and $\left(\mathcal{P}(\mathcal{Y})^{[0,T]}, \int_0^T \mathrm{TV}(\mu_t, \mu_t')\mathrm{d}t\right)$ are complete. Therefore, one can deduce that $\left(\mathcal{P}(\mathcal{X})^{[0,T]} \times \mathcal{P}(\mathcal{Y})^{[0,T]}, \mathcal{TV}^{[0,T]}\right)$ is also complete. We consider the Picard iteration mapping $\phi\left((\nu_t^{(n-1)}, \mu_t^{(n-1)})_{t\in[0,T]}\right) := (\nu_t^{(n)}, \mu_t^{(n)})_{t\in[0,T]}$ defined via (29) and (30) and show that $\phi$ admits a unique fixed point $(\nu_t, \mu_t)_{t\in[0,T]}$ in $\left(\mathcal{P}(\mathcal{X})^{[0,T]} \times \mathcal{P}(\mathcal{Y})^{[0,T]}, \mathcal{TV}^{[0,T]}\right)$, which is the solution to (19).

**Lemma A.2.** *The mapping $\phi$ admits a unique fixed point $(\nu_t, \mu_t)_{t\in[0,T]}$ in $\left(\mathcal{P}(\mathcal{X})^{[0,T]} \times \mathcal{P}(\mathcal{Y})^{[0,T]}, \mathcal{TV}^{[0,T]}\right)$.*

*Proof of Lemma A.2. Step 1: The sequence of flows $\left((\nu_t^{(n)}, \mu_t^{(n)})_{t\in[0,T]}\right)_{n=0}^{\infty}$ is a Cauchy sequence in $\left(\mathcal{P}(\mathcal{X})^{[0,T]} \times \mathcal{P}(\mathcal{Y})^{[0,T]}, \mathcal{TV}^{[0,T]}\right)$.*

From (29), we have

$$\begin{aligned}
\log \nu_t^{(n)}(x) - \log \nu_t^{(n-1)}(x) = -\int_0^t \frac{\sigma^2}{2} e^{-\frac{\sigma^2}{2}(t-s)} \times \\
\times \left[ \frac{2}{\sigma^2} \left( \frac{\delta F}{\delta \nu}(\nu_s^{(n-1)}, \mu_s^{(n-1)}, x) - \frac{\delta F}{\delta \nu}(\nu_s^{(n-2)}, \mu_s^{(n-2)}, x) \right) - \mathrm{D_{KL}}(\nu_s^{(n-1)}|\pi) + \mathrm{D_{KL}}(\nu_s^{(n-2)}|\pi) \right] \mathrm{d}s.
\end{aligned}$$

Multiplying both sides by $\nu_t^{(n)}(x)$ and integrating with respect to $x$, we obtain

$$
D_{\mathrm{KL}}(\nu_t^{(n)}|\nu_t^{(n-1)}) = -\int_0^t \frac{\sigma^2}{2} e^{-\frac{\sigma^2}{2}(t-s)} \left[ \frac{2}{\sigma^2} \int_{\mathcal{X}} \left( \frac{\delta F}{\delta \nu}(\nu_s^{(n-1)}, \mu_s^{(n-1)}, x) - \frac{\delta F}{\delta \nu}(\nu_s^{(n-2)}, \mu_s^{(n-2)}, x) \right) \nu_t^{(n)}(\mathrm{d}x) \right.
$$
$$
\left. - D_{\mathrm{KL}}(\nu_s^{(n-1)}|\pi) + D_{\mathrm{KL}}(\nu_s^{(n-2)}|\pi) \right] \mathrm{d}s. \quad (31)
$$

Moreover, note that

$$
\int_{\mathcal{X}} \left( \frac{\delta F}{\delta \nu}(\nu_s^{(n-1)}, \mu_s^{(n-1)}, x) - \frac{\delta F}{\delta \nu}(\nu_s^{(n-2)}, \mu_s^{(n-2)}, x) \right) \nu_t^{(n)}(\mathrm{d}x)
$$
$$
= \int_{\mathcal{X}} \left( \frac{\delta F}{\delta \nu}(\nu_s^{(n-1)}, \mu_s^{(n-1)}, x) - \frac{\delta F}{\delta \nu}(\nu_s^{(n-1)}, \mu_s^{(n-2)}, x) + \frac{\delta F}{\delta \nu}(\nu_s^{(n-1)}, \mu_s^{(n-2)}, x) - \frac{\delta F}{\delta \nu}(\nu_s^{(n-2)}, \mu_s^{(n-2)}, x) \right) \nu_t^{(n)}(\mathrm{d}x)
$$
$$
= \int_{\mathcal{X}} \int_{\mathcal{Y}} \int_0^1 \frac{\delta^2 F}{\delta \mu \delta \nu} \left( \nu_s^{(n-1)}, \mu_s^{(n-2)} + \lambda \left( \mu_s^{(n-1)} - \mu_s^{(n-2)} \right), x, w \right) \mathrm{d}\lambda \left( \mu_s^{(n-1)} - \mu_s^{(n-2)} \right)(\mathrm{d}w) \nu_t^{(n)}(\mathrm{d}x)
$$
$$
+ \int_{\mathcal{X}} \int_{\mathcal{X}} \int_0^1 \frac{\delta^2 F}{\delta \nu^2} \left( \nu_s^{(n-2)} + \lambda \left( \nu_s^{(n-1)} - \nu_s^{(n-2)} \right), \mu_s^{(n-2)}, x, z \right) \mathrm{d}\lambda \left( \nu_s^{(n-1)} - \nu_s^{(n-2)} \right)(\mathrm{d}z) \nu_t^{(n)}(\mathrm{d}x).
$$

Similarly, again from (29) we have

$$
\log \nu_t^{(n-1)}(x) - \log \nu_t^{(n)}(x) = -\int_0^t \frac{\sigma^2}{2} e^{-\frac{\sigma^2}{2}(t-s)} \times
$$
$$
\times \left[ \frac{2}{\sigma^2} \int_{\mathcal{X}} \left( \frac{\delta F}{\delta \nu}(\nu_s^{(n-2)}, \mu_s^{(n-2)}, x) - \frac{\delta F}{\delta \nu}(\nu_s^{(n-1)}, \mu_s^{(n-1)}, x) \right) \nu_t^{(n)}(\mathrm{d}x) \right.
$$
$$
\left. - D_{\mathrm{KL}}(\nu_s^{(n-2)}|\pi) + D_{\mathrm{KL}}(\nu_s^{(n-1)}|\pi) \right] \mathrm{d}s.
$$

Multiplying both sides by $\nu_t^{(n-1)}(x)$ and integrating with respect to $x$, we obtain

$$
D_{\mathrm{KL}}(\nu_t^{(n-1)}|\nu_t^{(n)}) = -\int_0^t \frac{\sigma^2}{2} e^{-\frac{\sigma^2}{2}(t-s)} \left[ \frac{2}{\sigma^2} \int_{\mathcal{X}} \left( \frac{\delta F}{\delta \nu}(\nu_s^{(n-2)}, \mu_s^{(n-2)}, x) - \frac{\delta F}{\delta \nu}(\nu_s^{(n-1)}, \mu_s^{(n-1)}, x) \right) \nu_t^{(n-1)}(\mathrm{d}x) \right.
$$
$$
\left. - D_{\mathrm{KL}}(\nu_s^{(n-2)}|\pi) + D_{\mathrm{KL}}(\nu_s^{(n-1)}|\pi) \right] \mathrm{d}s. \quad (32)
$$

Similarly as before, we note that

$$
\int_{\mathcal{X}} \left( \frac{\delta F}{\delta \nu}(\nu_s^{(n-2)}, \mu_s^{(n-2)}, x) - \frac{\delta F}{\delta \nu}(\nu_s^{(n-1)}, \mu_s^{(n-1)}, x) \right) \nu_t^{(n-1)}(\mathrm{d}x)
$$
$$
= -\int_{\mathcal{X}} \left( \frac{\delta F}{\delta \nu}(\nu_s^{(n-1)}, \mu_s^{(n-1)}, x) - \frac{\delta F}{\delta \nu}(\nu_s^{(n-1)}, \mu_s^{(n-2)}, x) + \frac{\delta F}{\delta \nu}(\nu_s^{(n-1)}, \mu_s^{(n-2)}, x) - \frac{\delta F}{\delta \nu}(\nu_s^{(n-2)}, \mu_s^{(n-2)}, x) \right) \nu_t^{(n-1)}(\mathrm{d}x)
$$
$$
= -\int_{\mathcal{X}} \int_{\mathcal{Y}} \int_0^1 \frac{\delta^2 F}{\delta \mu \delta \nu} \left( \nu_s^{(n-1)}, \mu_s^{(n-2)} + \lambda \left( \mu_s^{(n-1)} - \mu_s^{(n-2)} \right), x, w \right) \times \mathrm{d}\lambda \left( \mu_s^{(n-1)} - \mu_s^{(n-2)} \right)(\mathrm{d}w) \nu_t^{(n-1)}(\mathrm{d}x)
$$
$$
- \int_{\mathcal{X}} \int_{\mathcal{X}} \int_0^1 \frac{\delta^2 F}{\delta \nu^2} \left( \nu_s^{(n-2)} + \lambda \left( \nu_s^{(n-1)} - \nu_s^{(n-2)} \right), \mu_s^{(n-2)}, x, z \right) \times \mathrm{d}\lambda \left( \nu_s^{(n-1)} - \nu_s^{(n-2)} \right)(\mathrm{d}z) \nu_t^{(n-1)}(\mathrm{d}x).
$$

Combining (31) and (32), we obtain

$$
D_{KL}(\nu_t^{(n)}|\nu_t^{(n-1)}) + D_{KL}(\nu_t^{(n-1)}|\nu_t^{(n)}) = -\int_0^t e^{-\frac{\sigma^2}{2}(t-s)} \times \Bigg[
$$

$$
\int_{\mathcal{X}}\int_{\mathcal{Y}}\int_0^1 \frac{\delta^2 F}{\delta\mu\delta\nu}\left(\nu_s^{(n-1)}, \mu_s^{(n-2)} + \lambda\left(\mu_s^{(n-1)} - \mu_s^{(n-2)}\right), x, w\right) d\lambda \left(\mu_s^{(n-1)} - \mu_s^{(n-2)}\right)(dw)\left(\nu_t^{(n)} - \nu_t^{(n-1)}\right)(dx)
$$

$$
+\int_{\mathcal{X}}\int_{\mathcal{X}}\int_0^1 \frac{\delta^2 F}{\delta\nu^2}\left(\nu_s^{(n-2)} + \lambda\left(\nu_s^{(n-1)} - \nu_s^{(n-2)}\right), \mu_s^{(n-2)}, x, z\right) d\lambda \left(\nu_s^{(n-1)} - \nu_s^{(n-2)}\right)(dz)\left(\nu_t^{(n)} - \nu_t^{(n-1)}\right)(dx)\Bigg] ds.
$$

Hence, due to Assumption 3, we get

$$
D_{KL}(\nu_t^{(n)}|\nu_t^{(n-1)}) + D_{KL}(\nu_t^{(n-1)}|\nu_t^{(n)})
$$

$$
\leq TV(\nu_t^{(n)}, \nu_t^{(n-1)})\int_0^t e^{-\frac{\sigma^2}{2}(t-s)}\left(C_{\mu,\nu} TV(\mu_s^{(n-1)}, \mu_s^{(n-2)}) + C_{\nu,\nu} TV(\nu_s^{(n-1)}, \nu_s^{(n-2)})\right) ds
$$

$$
\leq \max\{C_{\mu,\nu}, C_{\nu,\nu}\} TV(\nu_t^{(n)}, \nu_t^{(n-1)})\int_0^t e^{-\frac{\sigma^2}{2}(t-s)}\left(TV(\mu_s^{(n-1)}, \mu_s^{(n-2)}) + TV(\nu_s^{(n-1)}, \nu_s^{(n-2)})\right) ds.
$$

By the Pinsker-Csizsar inequality, $TV^2(\nu_t^{(n)}, \nu_t^{(n-1)}) \leq \frac{1}{2} D_{KL}(\nu_t^{(n)}|\nu_t^{(n-1)})$, and hence

$$
4\,TV^2(\nu_t^{(n)}, \nu_t^{(n-1)}) \leq \max\{C_{\mu,\nu}, C_{\nu,\nu}\} TV(\nu_t^{(n)}, \nu_t^{(n-1)})\int_0^t e^{-\frac{\sigma^2}{2}(t-s)}\Big(TV(\mu_s^{(n-1)}, \mu_s^{(n-2)})
$$

$$
+ TV(\nu_s^{(n-1)}, \nu_s^{(n-2)})\Big)ds,
$$

which gives

$$
TV(\nu_t^{(n)}, \nu_t^{(n-1)}) \leq \frac{1}{4}\max\{C_{\mu,\nu}, C_{\nu,\nu}\}\int_0^t e^{-\frac{\sigma^2}{2}(t-s)}\Big(TV(\mu_s^{(n-1)}, \mu_s^{(n-2)}) + TV(\nu_s^{(n-1)}, \nu_s^{(n-2)})\Big)ds.
$$

An almost identical argument leads to

$$
TV(\mu_t^{(n)}, \mu_t^{(n-1)}) \leq \frac{1}{4}\max\{C_{\nu,\mu}, C_{\mu,\mu}\}\int_0^t e^{-\frac{\sigma^2}{2}(t-s)}\Big(TV(\mu_s^{(n-1)}, \mu_s^{(n-2)}) + TV(\nu_s^{(n-1)}, \nu_s^{(n-2)})\Big)ds.
$$

If we set $C_{\max} := \max\{C_{\mu,\nu}, C_{\nu,\nu}\} + \max\{C_{\nu,\mu}, C_{\mu,\mu}\}$, and add the previous two inequalities, we obtain

$$
TV(\nu_t^{(n)}, \nu_t^{(n-1)}) + TV(\mu_t^{(n)}, \mu_t^{(n-1)}) \leq \frac{C_{\max}}{4}\int_0^t e^{-\frac{\sigma^2}{2}(t-s)}\Big(TV(\mu_s^{(n-1)}, \mu_s^{(n-2)}) + TV(\nu_s^{(n-1)}, \nu_s^{(n-2)})\Big)ds.
$$

$$
\leq \left(\frac{C_{\max}}{4}\right)^{n-1} e^{-\frac{\sigma^2}{2}t}\int_0^t \int_0^{t_1}\ldots\int_0^{t_{n-2}} e^{\frac{\sigma^2}{2}t_{n-1}}\Big(TV(\nu_{t_{n-1}}^{(1)}, \nu_{t_{n-1}}^{(0)}) + TV(\mu_{t_{n-1}}^{(1)}, \mu_{t_{n-1}}^{(0)})\Big)dt_{n-1}\ldots dt_2 dt_1
$$

$$
\leq \left(\frac{C_{\max}}{4}\right)^{n-1} e^{-\frac{\sigma^2}{2}t}\frac{t^{n-2}}{(n-2)!}\int_0^t e^{\frac{\sigma^2}{2}t_{n-1}}\Big(TV(\nu_{t_{n-1}}^{(1)}, \nu_{t_{n-1}}^{(0)}) + TV(\mu_{t_{n-1}}^{(1)}, \mu_{t_{n-1}}^{(0)})\Big)dt_{n-1}
$$

$$
\leq \left(\frac{C_{\max}}{4}\right)^{n-1}\frac{t^{n-2}}{(n-2)!}\int_0^t \Big(TV(\nu_{t_{n-1}}^{(1)}, \nu_{t_{n-1}}^{(0)}) + TV(\mu_{t_{n-1}}^{(1)}, \mu_{t_{n-1}}^{(0)})\Big)dt_{n-1},
$$

where in the third inequality we bounded $\int_0^{t_{n-2}} dt_{n-1} \leq \int_0^t dt_{n-1}$ and in the fourth inequality we bounded $e^{\frac{\sigma^2}{2}t_{n-1}} \leq e^{\frac{\sigma^2}{2}t}$. Hence, we obtain

$$
\int_0^T TV(\nu_t^{(n)}, \nu_t^{(n-1)})dt + \int_0^T TV(\mu_t^{(n)}, \mu_t^{(n-1)})dt
$$

$$
\leq \left(\frac{C_{\max}}{4}\right)^{n-1}\frac{T^{n-1}}{(n-1)!}\left(\int_0^T TV(\nu_{t_{n-1}}^{(1)}, \nu_{t_{n-1}}^{(0)})dt_{n-1} + \int_0^T TV(\mu_{t_{n-1}}^{(1)}, \mu_{t_{n-1}}^{(0)})dt_{n-1}\right).
$$

Using the definition of $\mathcal{TV}^{[0,T]}$, the last inequality becomes

$$\mathcal{TV}^{[0,T]}\left((\nu_t^{(n)},\mu_t^{(n)})_{t\in[0,T]},(\nu_t^{(n-1)},\mu_t^{(n-1)})_{t\in[0,T]}\right)$$
$$\leq\left(\frac{C_{\max}}{4}\right)^{n-1}\frac{T^{n-1}}{(n-1)!}\mathcal{TV}^{[0,T]}\left((\nu_t^{(1)},\mu_t^{(1)})_{t\in[0,T]},(\nu_t^{(0)},\mu_t^{(0)})_{t\in[0,T]}\right).$$

By choosing $n$ sufficiently large, we conclude that $\left((\nu_t^{(n)},\mu_t^{(n)})_{t\in[0,T]}\right)_{n=0}^{\infty}$ is a Cauchy sequence. By completeness of $\left(\mathcal{P}(\mathcal{X})^{[0,T]}\times\mathcal{P}(\mathcal{Y})^{[0,T]},\mathcal{TV}^{[0,T]}\right)$, the sequence admits a limit point $(\nu_t,\mu_t)_{t\in[0,T]}\in\left(\mathcal{P}(\mathcal{X})^{[0,T]}\times\mathcal{P}(\mathcal{Y})^{[0,T]},\mathcal{TV}^{[0,T]}\right)$.

*Step 2: The limit point $(\nu_t,\mu_t)_{t\in[0,T]}$ is a fixed point of $\phi$.* From Step 1, we obtain that for Lebesgue-almost all $t\in[0,T]$ we have

$$\mathrm{TV}(\nu_t^{(n)},\nu_t)\to 0,\quad \mathrm{TV}(\mu_t^{(n)},\mu_t)\to 0\qquad\text{as }n\to\infty.$$

Thus, by (12) and (13), we have that for Lebesgue-almost all $t\in[0,T]$ and any fixed $(x,y)\in\mathcal{X}\times\mathcal{Y}$, the flat derivatives $\frac{\delta F}{\delta\nu}(\nu_t^{(n-1)},\mu_t^{(n-1)},x)$ and $\frac{\delta F}{\delta\mu}(\nu_t^{(n-1)},\mu_t^{(n-1)},y)$ from (29) and (30) converge to $\frac{\delta F}{\delta\nu}(\nu_t,\mu_t,x)$ and $\frac{\delta F}{\delta\mu}(\nu_t,\mu_t,y)$, respectively, as $n\to\infty$. Using Assumption 2, we can apply the dominated convergence theorem and obtain

$$\lim_{n\to\infty}\int_0^t\frac{\delta F}{\delta\nu}(\nu_s^{(n-1)},\mu_s^{(n-1)},x)\mathrm{d}s=\int_0^t\frac{\delta F}{\delta\nu}(\nu_s,\mu_s,x)\mathrm{d}s,$$

$$\lim_{n\to\infty}\int_0^t\frac{\delta F}{\delta\mu}(\nu_s^{(n-1)},\mu_s^{(n-1)},y)\mathrm{d}s=\int_0^t\frac{\delta F}{\delta\mu}(\nu_s,\mu_s,y)\mathrm{d}s,$$

for Lebesgue-almost all $t\in[0,T]$ and any fixed $(x,y)\in\mathcal{X}\times\mathcal{Y}$.

Moreover, in (29) and (30), using Assumption 2, (15), Lemma A.1 and the fact that $(\nu_t^{(n)},\mu_t^{(n)})_{t\in[0,T]}$ is a flow of probability measures for all $n$, we obtain for all $n$ and $t\in[0,T]$, and any fixed $(x,y)\in\mathcal{X}\times\mathcal{Y}$, that

$$\nu_t^{(n)}(x)\log\frac{\nu_t^{(n)}(x)}{\pi(x)}\leq 3\log R_\nu+\frac{6}{\sigma^2}C_\nu,$$

$$\mu_t^{(n)}(y)\log\frac{\mu_t^{(n)}(y)}{\rho(y)}\leq 3\log R_\mu+\frac{6}{\sigma^2}C_\mu.$$

Furthermore, since $(\nu_t^{(n)},\mu_t^{(n)})_{t\in[0,T]}$ are considered to be absolutely continuous with respect to the Lebesgue measure for all $n$, it follows that convergence in TV distance implies convergence in $L^1$. Then due to the dominated convergence theorem, we deduce that $\mathrm{D_{KL}}(\nu_t^{(n)}|\pi)$ and $\mathrm{D_{KL}}(\mu_t^{(n)}|\rho)$ converge to $\mathrm{D_{KL}}(\nu_t|\pi)$ and $\mathrm{D_{KL}}(\mu_t|\rho)$, respectively, for Lebesgue-almost all $t\in[0,T]$, as $n\to\infty$. Hence, using again Lemma A.1 and the dominated convergence theorem, we obtain

$$\lim_{n\to\infty}\int_0^t\mathrm{D_{KL}}(\nu_s^{(n)}|\pi)\mathrm{d}s=\int_0^t\mathrm{D_{KL}}(\nu_s|\pi)\mathrm{d}s,$$

$$\lim_{n\to\infty}\int_0^t\mathrm{D_{KL}}(\mu_s^{(n)}|\rho)\mathrm{d}s=\int_0^t\mathrm{D_{KL}}(\mu_s|\rho)\mathrm{d}s,$$

for Lebesgue-almost all $t\in[0,T]$.

Therefore, letting $n\to\infty$ in (29) and (30) and using the fact that $x\mapsto\log x$ is continuous on $(0,\infty)$, we conclude that $(\nu_t,\mu_t)_{t\in[0,T]}$ is a fixed point of $\phi$.

*Step 3: The fixed point $(\nu_t,\mu_t)_{t\in[0,T]}$ of $\phi$ is unique.* Suppose, for the contrary, that $\phi$ admits two fixed points $(\nu_t,\mu_t)_{t\in[0,T]}$ and $(\bar{\nu}_t,\bar{\mu}_t)_{t\in[0,T]}$ such that $\nu_0=\bar{\nu}_0$ and $\mu_0=\bar{\mu}_0$. Then repeating the same calculations from Step 1, we arrive at

$$\mathrm{D_{KL}}(\nu_t|\bar{\nu}_t)+\mathrm{D_{KL}}(\bar{\nu}_t|\nu_t)\leq\max\{C_{\mu,\nu},C_{\nu,\nu}\}\mathrm{TV}(\nu_t,\bar{\nu}_t)\int_0^t e^{-\frac{\sigma^2}{2}(t-s)}\left(\mathrm{TV}(\mu_s,\bar{\mu}_s)+\mathrm{TV}(\nu_s,\bar{\nu}_s)\right)\mathrm{d}s.$$

By the Pinsker-Csizsar inequality, $\mathrm{TV}^2(\nu_t, \bar{\nu}_t) \leq \frac{1}{2}\, \mathrm{D_{KL}}(\nu_t|\bar{\nu}_t)$, and hence

$$4\, \mathrm{TV}^2(\nu_t, \bar{\nu}_t) \leq \max\{C_{\mu,\nu}, C_{\nu,\nu}\}\, \mathrm{TV}(\nu_t, \bar{\nu}_t) \int_0^t e^{-\frac{\sigma^2}{2}(t-s)} \left(\mathrm{TV}(\mu_s, \bar{\mu}_s) + \mathrm{TV}(\nu_s, \bar{\nu}_s)\right) \mathrm{d}s,$$

which gives

$$\mathrm{TV}(\nu_t, \bar{\nu}_t) \leq \frac{1}{4} \max\{C_{\mu,\nu}, C_{\nu,\nu}\} \int_0^t e^{-\frac{\sigma^2}{2}(t-s)} \left(\mathrm{TV}(\mu_s, \bar{\mu}_s) + \mathrm{TV}(\nu_s, \bar{\nu}_s)\right) \mathrm{d}s,$$

An almost identical argument leads to

$$\mathrm{TV}(\mu_t, \bar{\mu}_t) \leq \frac{1}{4} \max\{C_{\nu,\mu}, C_{\mu,\mu}\} \int_0^t e^{-\frac{\sigma^2}{2}(t-s)} \left(\mathrm{TV}(\mu_s, \bar{\mu}_s) + \mathrm{TV}(\nu_s, \bar{\nu}_s)\right) \mathrm{d}s.$$

If we set $C_{\max} := \max\{C_{\mu,\nu}, C_{\nu,\nu}\} + \max\{C_{\nu,\mu}, C_{\mu,\mu}\}$, and add the previous two inequalities, we obtain

$$\mathrm{TV}(\nu_t, \bar{\nu}_t) + \mathrm{TV}(\mu_t, \bar{\mu}_t) \leq \frac{C_{\max}}{4} \int_0^t e^{-\frac{\sigma^2}{2}(t-s)} \left(\mathrm{TV}(\mu_s, \bar{\mu}_s) + \mathrm{TV}(\nu_s, \bar{\nu}_s)\right) \mathrm{d}s.$$

For each $t \in [0, T]$, denote $f(t) := \int_0^t e^{\frac{\sigma^2}{2}s} \left(\mathrm{TV}(\mu_s, \bar{\mu}_s) + \mathrm{TV}(\nu_s, \bar{\nu}_s)\right) \mathrm{d}s$. Observe that $f \geq 0$ and $f(0) = 0$. Then, by Gronwall's lemma, we obtain

$$0 \leq f(t) \leq e^{\frac{C_{\max}}{4}} f(0) = 0,$$

and hence

$$\mathrm{TV}(\nu_t, \bar{\nu}_t) + \mathrm{TV}(\mu_t, \bar{\mu}_t) = 0,$$

for Lebesgue-almost all $t \in [0, T]$, which implies

$$\nu_t = \bar{\nu}_t, \quad \mu_t = \bar{\mu}_t,$$

for Lebesgue-almost all $t \in [0, T]$. Therefore, the fixed point $(\nu_t, \mu_t)_{t \in [0,T]}$ of $\phi$ must be unique.

From Steps 1, 2 and 3, we obtain the existence and uniqueness of a pair of flows $(\nu_t, \mu_t)_{t \in [0,T]}$ satisfying (27) for any $T > 0$. $\qquad\square$

Returning to the proof of Theorem 2.2, recall from Step 1 of Lemma A.2 that for Lebesgue-almost all $t \in [0, T]$ we have

$$\mathrm{TV}(\nu_t^{(n)}, \nu_t) \to 0, \quad \mathrm{TV}(\mu_t^{(n)}, \mu_t) \to 0 \qquad \text{as } n \to \infty,$$

which implies

$$\nu_t^{(n)} \to \nu_t, \quad \mu_t^{(n)} \to \mu_t \qquad \text{weakly as } n \to \infty.$$

Hence, using the lower semi-continuity of the entropy, we obtain

$$\mathrm{D_{KL}}(\nu_t|\pi) \leq \liminf_{n\to\infty} \mathrm{D_{KL}}(\nu_t^{(n)}|\pi) \leq 2 \log R_\nu + \frac{4}{\sigma^2} C_\nu, \tag{33}$$

$$\mathrm{D_{KL}}(\mu_t|\rho) \leq \liminf_{n\to\infty} \mathrm{D_{KL}}(\mu_t^{(n)}|\rho) \leq 2 \log R_\mu + \frac{4}{\sigma^2} C_\mu, \tag{34}$$

where both second inequalities follow from Lemma A.1. In order to ensure that the solution $(\nu_t, \mu_t)_{t \in [0,T]}$ can be extended to all $t \geq 0$, we first need to prove the bound on the ratios $\nu_t/\pi$ and $\mu_t/\rho$ in (21).

*Step 2: Ratio condition* (21). Using (29) and (30), we see that for any $t \in [0, T]$ we have

$$\log \frac{\nu_t^{(n)}(x)}{\pi(x)} = e^{-\frac{\sigma^2}{2}t} \log \frac{\nu_0(x)}{\pi(x)} - \int_0^t \frac{\sigma^2}{2} e^{-\frac{\sigma^2}{2}(t-s)} \left(\frac{2}{\sigma^2} \frac{\delta F}{\delta \nu}(\nu_s^{(n-1)}, \mu_s^{(n-1)}, x) - \mathrm{D_{KL}}(\nu_s^{(n-1)}|\pi)\right) \mathrm{d}s,$$

$$\log \frac{\mu_t^{(n)}(y)}{\rho(y)} = e^{-\frac{\sigma^2}{2}t} \log \frac{\mu_0(y)}{\rho(y)} + \int_0^t \frac{\sigma^2}{2} e^{-\frac{\sigma^2}{2}(t-s)} \left(\frac{2}{\sigma^2} \frac{\delta F}{\delta \mu}(\nu_s^{(n-1)}, \mu_s^{(n-1)}, y) + \mathrm{D_{KL}}(\mu_s^{(n-1)}|\rho)\right) \mathrm{d}s.$$

Using Assumption 2, (15), (33) and (34) we obtain

$$\log \frac{\nu_t(x)}{\pi(x)} \le 3 \log R_\nu + \frac{6}{\sigma^2} C_\nu,$$

$$\log \frac{\mu_t(y)}{\rho(y)} \le 3 \log R_\mu + \frac{6}{\sigma^2} C_\mu.$$

Hence we can choose $R_{1,\nu} := 1 + \exp\left(3 \log R_\nu + \frac{6}{\sigma^2} C_\nu\right)$ and $R_{1,\mu} := 1 + \exp\left(3 \log R_\mu + \frac{6}{\sigma^2} C_\mu\right)$. Note $R_{1,\nu}, R_{1,\mu} > 1$ are conveniently chosen so that $\log R_{1,\nu}, \log R_{1,\mu} > 0$ in our subsequent calculations. Obtaining a lower bound on $\frac{\nu_t(x)}{\pi(x)}$ and $\frac{\mu_t(y)}{\rho(y)}$ follows similarly, by using (14) instead of (15).

*Step 3: Existence of the gradient flow on $[0, \infty)$.* In order to complete our proof, note that the unique solution $(\nu_t, \mu_t)_{t \in [0,T]}$ to (27) can also be expressed as

$$
\begin{aligned}
\nu_t(x) &= \nu_0(x) \exp\left(-\int_0^t \left(\frac{\delta F}{\delta \nu}(\nu_s, \mu_s, x) + \frac{\sigma^2}{2} \log\left(\frac{\nu_s(x)}{\pi(x)}\right) - \frac{\sigma^2}{2} \mathrm{D}_{\mathrm{KL}}(\nu_s | \pi)\right) \mathrm{d}s\right), \\
\mu_t(y) &= \mu_0(y) \exp\left(\int_0^t \left(\frac{\delta F}{\delta \mu}(\nu_s, \mu_s, y) - \frac{\sigma^2}{2} \log\left(\frac{\mu_s(y)}{\rho(y)}\right) + \frac{\sigma^2}{2} \mathrm{D}_{\mathrm{KL}}(\mu_s | \rho)\right) \mathrm{d}s\right).
\end{aligned}
\tag{35}
$$

Then it follows that the flows $(\nu_t, \mu_t)_{t \in [0,T]}$ are continuous and differentiable in time.

From 2, (33), (34) and (21), we obtain for any $t \in [0, T]$

$$
\left| \frac{\delta F}{\delta \nu}(\nu_t, \mu_t, x) + \frac{\sigma^2}{2} \log\left(\frac{\nu_t(x)}{\pi(x)}\right) - \frac{\sigma^2}{2} \mathrm{D}_{\mathrm{KL}}(\nu_t | \pi) \right|
$$
$$
\le 3 C_\nu + \frac{\sigma^2}{2} \left(\max\{|\log r_{1,\nu}|, \log R_{1,\nu}\} + 2 \log R_\nu\right) =: C_{V,\nu}, \quad (36)
$$

$$
\left| \frac{\delta F}{\delta \mu}(\nu_t, \mu_t, y) - \frac{\sigma^2}{2} \log\left(\frac{\mu_t(y)}{\rho(y)}\right) + \frac{\sigma^2}{2} \mathrm{D}_{\mathrm{KL}}(\mu_t | \rho) \right|
$$
$$
\le 3 C_\mu + \frac{\sigma^2}{2} \left(\max\{|\log r_{1,\mu}|, \log R_{1,\mu}\} + 2 \log R_\mu\right) =: C_{V,\mu}.
$$

This gives $\|\nu_t\|_{TV} \le \|\nu_0\|_{TV} e^{C_{V,\nu} t}$ and $\|\mu_t\|_{TV} \le \|\mu_0\|_{TV} e^{C_{V,\mu} t}$, and shows that $\nu_t$ and $\mu_t$ do not explode in any finite time, hence we obtain a global solution $(\nu_t, \mu_t)_{t \in [0,\infty)}$, which is continuous and differentiable in time. In particular, the bounds in (33), (34), (21) and (22) hold for all $t > 0$.

$\square$

# B    Notation and definitions

In this section we recall some important definitions. Following Carmona & Delarue (2018, Definition 5.43), we start with the notion of differentiability on the space of probability measure that we utilize throughout the paper.

**Definition B.1.** *Fix $p \ge 0$. For any $\mathcal{M} \subseteq \mathbb{R}^d$, let $\mathcal{P}_p(\mathcal{M})$ be the space of probability measures on $\mathcal{M}$ with finite $p$-th moments. A function $F : \mathcal{P}_p(\mathcal{M}) \to \mathbb{R}$ admits first-order flat derivative on $\mathcal{P}_p(\mathcal{M})$, if there exists a function $\frac{\delta F}{\delta \nu} : \mathcal{P}_p(\mathcal{M}) \times \mathcal{M} \to \mathbb{R}$, such that*

1. *the map $\mathcal{P}_p(\mathcal{M}) \times \mathcal{M} \ni (m, x) \mapsto \frac{\delta F}{\delta m}(m, x)$ is jointly continuous with respect to the product topology, where $\mathcal{P}_p(\mathcal{M})$ is endowed with the weak topology,*

2. *For any $m \in \mathcal{P}_p(\mathcal{M})$, there exists $C > 0$ such that, for all $x \in \mathcal{M}$, we have*

$$
\left| \frac{\delta F}{\delta m}(m, x) \right| \le C \left(1 + |x|^p\right),
$$

3. For all $m, m' \in \mathcal{P}_p(\mathcal{M})$, it holds that

$$F(m') - F(m) = \int_0^1 \int_{\mathcal{M}} \frac{\delta F}{\delta m}(m + \varepsilon(m' - m), x)(m' - m)(\mathrm{d}x)\mathrm{d}\varepsilon. \tag{37}$$

The functional $\frac{\delta F}{\delta m}$ is then called the flat derivative of $F$ on $\mathcal{P}_p(\mathcal{M})$. We note that $\frac{\delta F}{\delta m}$ exists up to an additive constant, and thus we make the normalizing convention $\int_{\mathcal{M}} \frac{\delta F}{\delta m}(m, x)m(\mathrm{d}x) = 0$.

If, for any fixed $x \in \mathcal{M}$, the map $m \mapsto \frac{\delta F}{\delta m}(m, x)$ satisfies Definition B.1, we say that $F$ admits a second-order flat derivative denoted by $\frac{\delta^2 F}{\delta m^2}$. Consequently, by Definition B.1, there exists a functional $\frac{\delta^2 F}{\delta m^2} : \mathcal{P}_p(\mathcal{M}) \times \mathcal{M} \times \mathcal{M} \to \mathbb{R}$ such that

$$\frac{\delta F}{\delta m}(m', x) - \frac{\delta F}{\delta m}(m, x) = \int_0^1 \int_{\mathcal{M}} \frac{\delta^2 F}{\delta m^2}(\nu + \varepsilon(m' - m), x, x')(m' - m)(\mathrm{d}x')\mathrm{d}\varepsilon. \tag{38}$$

**Example B.2.** *We provide some simple examples of functions $F : \mathcal{P}_p(\mathcal{M}) \to \mathbb{R}$ and their flat derivatives.*

1. Let $F(m) = \int_{\mathcal{M}} f(x)m(\mathrm{d}x)$, where $f : \mathcal{M} \to \mathbb{R}$ is continuous and has p-th order growth. Then

$$\frac{\delta F}{\delta m}(m, x) = f(x), \quad \frac{\delta^2 F}{\delta m^2}(m, x) = 0.$$

2. Let $F(\nu, \mu) = \int_{\mathcal{M}} \int_{\mathcal{M}} f(x, y)\nu(\mathrm{d}x)\mu(\mathrm{d}y)$. Then

$$\frac{\delta F}{\delta \nu}(\nu, \mu, x) = \int_{\mathcal{M}} f(x, y)\mu(\mathrm{d}y), \quad \frac{\delta F}{\delta \mu}(\mu, \nu, y) = \int_{\mathcal{M}} f(x, y)\nu(\mathrm{d}x), \quad \frac{\delta^2 F}{\delta \nu^2}(\nu, \mu, x) = \frac{\delta^2 F}{\delta \mu^2}(\mu, \nu, y) = 0.$$

3. Let $F(\nu, \mu) = g\left(\int_{\mathcal{M}} \int_{\mathcal{M}} f(x, y)\nu(\mathrm{d}x)\mu(\mathrm{d}y)\right)$ with $f$ continuous and $g$ twice continuously differentiable. Then using the chain rule we obtain

$$\frac{\delta F}{\delta \nu}(\nu, \mu, x) = g'\left(\int_{\mathcal{M}} \int_{\mathcal{M}} f(x, y)\nu(\mathrm{d}x)\mu(\mathrm{d}y)\right) \int_{\mathcal{M}} f(x, y)\mu(\mathrm{d}y),$$

$$\frac{\delta F}{\delta \mu}(\mu, \nu, y) = g'\left(\int_{\mathcal{M}} \int_{\mathcal{M}} f(x, y)\nu(\mathrm{d}x)\mu(\mathrm{d}y)\right) \int_{\mathcal{M}} f(x, y)\nu(\mathrm{d}x),$$

$$\frac{\delta^2 F}{\delta \mu \delta \nu}(\mu, \nu, x, y) = \frac{\delta^2 F}{\delta \nu \delta \mu}(\nu, \mu, y, x) = g'\left(\int_{\mathcal{M}} \int_{\mathcal{M}} f(x, y)\nu(\mathrm{d}x)\mu(\mathrm{d}y)\right) f(x, y)$$
$$+ \int_{\mathcal{M}} f(x, y)\mu(\mathrm{d}y)g''\left(\int_{\mathcal{M}} \int_{\mathcal{M}} f(x, y)\nu(\mathrm{d}x)\mu(\mathrm{d}y)\right) \int_{\mathcal{M}} f(x, y)\nu(\mathrm{d}x),$$

$$\frac{\delta^2 F}{\delta \nu^2}(\nu, \mu, x, x') = g''\left(\int_{\mathcal{M}} \int_{\mathcal{M}} f(x, y)\nu(\mathrm{d}x)\mu(\mathrm{d}y)\right) \int_{\mathcal{M}} f(x, y)\mu(\mathrm{d}y) \int_{\mathcal{M}} f(x', y)\mu(\mathrm{d}y),$$

$$\frac{\delta^2 F}{\delta \mu^2}(\nu, \mu, y, y') = g''\left(\int_{\mathcal{M}} \int_{\mathcal{M}} f(x, y)\nu(\mathrm{d}x)\mu(\mathrm{d}y)\right) \int_{\mathcal{M}} f(x, y)\nu(\mathrm{d}x) \int_{\mathcal{M}} f(x, y')\nu(\mathrm{d}x).$$

**Definition B.3** (TV distance between probability measures; (Tsybakov, 2008), Definition 2.4). *Let $(\mathcal{M}, \mathcal{A})$ be a measurable space and let $P$ and $Q$ be probability measures on $(\mathcal{M}, \mathcal{A})$. Assume that $\mu$ is a $\sigma$-finite measure on $(\mathcal{M}, \mathcal{A})$ such that $P$ and $Q$ are absolutely continuous with respect to $\mu$ and let $p$ and $q$ denote their probability density functions, respectively. The total variation distance between $P$ and $Q$ is defined as:*

$$\mathrm{TV}(P, Q) := \sup_{A \in \mathcal{A}} |P(A) - Q(A)| = \sup_{A \in \mathcal{A}} \left| \int_A (p - q)\mathrm{d}\mu \right|.$$

## C  A formal derivation of the Fisher-Rao gradient flow

In this section, we follow Lu et al. (2019); Yan et al. (2023); Gallouët & Monsaingeon (2017) to formally derive the Fisher-Rao gradient flow of a function $G : \mathcal{P}_{\mathrm{ac}}(\mathcal{M}) \to \mathbb{R}$, with $\mathcal{M} \subseteq \mathbb{R}$. Following Otto (2001), we define the tangent space $T_\lambda^{\mathrm{FR}}(\mathcal{M})$ at any $\lambda \in \mathcal{P}_{\mathrm{ac}}(\mathcal{M})$ by

$$T_\lambda^{\mathrm{FR}}(\mathcal{M}) := \left\{ \text{functions } \xi \text{ on } \mathcal{M} \text{ such that } \int_{\mathcal{M}} \xi(x)\mathrm{d}x = 0 \right\}.$$

The tangent space $T_\lambda^{\mathrm{FR}}(\mathcal{M})$ can also be identified with

$$T_\lambda^{\mathrm{FR}}(\mathcal{M}) \left\{ \xi : \xi(x) = \lambda(x) \left( r(x) - \int_{\mathcal{M}} r(x)\lambda(x)\mathrm{d}x \right), \text{ with } r \in L^2(\mathcal{M}; \lambda) \right\}.$$

On this tangent space, we define the Riemannian metric $g_\lambda^{\mathrm{FR}} : T_\lambda^{\mathrm{FR}}(\mathcal{M}) \times T_\lambda^{\mathrm{FR}}(\mathcal{M}) \to \mathbb{R}$ at $\lambda$ by

$$g_\lambda^{\mathrm{FR}}(\xi_1, \xi_2) := \int_{\mathcal{M}} \frac{\xi_1(x) \cdot \xi_2(x)}{\lambda(x)} \mathrm{d}x, \quad \text{for } \xi_1, \xi_2 \in T_\lambda^{\mathrm{FR}}(\mathcal{M}). \tag{39}$$

Then for any $\xi_i(x) = \lambda(x)\left( r_i(x) - \int_{\mathcal{M}} r_i(x)\lambda(x)\mathrm{d}x \right)$, for $i = 1, 2$, the metric $g_\lambda^{\mathrm{FR}}$ becomes

$$g_\lambda^{\mathrm{FR}}(\xi_1, \xi_2) = \int_{\mathcal{M}} r_1(x)r_2(x)\lambda(x)\mathrm{d}x - \int_{\mathcal{M}} r_1(x)\lambda(x)\mathrm{d}x \int_{\mathcal{M}} r_2(x)\lambda(x)\mathrm{d}x.$$

With the metric $g_\lambda^{\mathrm{FR}}$, the Fisher-Rao distance FR, defined by (3), can be viewed as the geodesic distance on $\left( \mathcal{P}_{\mathrm{ac}}(\mathcal{M}), g_\lambda^{\mathrm{FR}} \right)$.

Now, we derive the Fisher-Rao gradient flow for a given function $G : \mathcal{P}_{\mathrm{ac}}(\mathcal{M}) \to \mathbb{R}$, which we assume is flat differentiable (cf. Definition B.1). Let $(\lambda_t)_{t \geq 0}$ be a smooth curve starting from $\lambda_0 = \lambda \in \mathcal{P}_{\mathrm{ac}}(\mathcal{M})$ with arbitrary tangent vector

$$\xi(x) = \partial_t \lambda_t(x)|_{t=0} = \lambda(x) \left( r(x) - \int_{\mathcal{M}} r(x)\lambda(x)\mathrm{d}x \right). \tag{40}$$

The metric gradient $\mathrm{grad}^{\mathrm{FR}} G \in L^2(\mathcal{M}; \lambda)$ of $G$ with respect to $g_\lambda^{\mathrm{FR}}$ is defined by

$$\left. \frac{\mathrm{d}}{\mathrm{d}t} G(\lambda_t) \right|_{t=0} = g_\lambda^{\mathrm{FR}} \left( \mathrm{grad}^{\mathrm{FR}} G(\lambda), \xi \right), \tag{41}$$

and the gradient flow of $G$ on $(\mathcal{P}_{\mathrm{ac}}(\mathcal{M}), \mathrm{FR})$ is defined by

$$\partial_t \lambda_t(x) = -\mathrm{grad}^{\mathrm{FR}} G(\lambda_t)(x). \tag{42}$$

Then, by the chain rule, (40) and (39), we obtain

$$\left. \frac{\mathrm{d}}{\mathrm{d}t} G(\lambda_t) \right|_{t=0} = \int_{\mathcal{M}} \frac{\delta G}{\delta \lambda}(\lambda, x)[\partial_t \lambda_t(x)]|_{t=0} \mathrm{d}x = \int_{\mathcal{M}} \frac{\delta G}{\delta \lambda}(\lambda, x)\xi(x)\mathrm{d}x = g_\lambda^{\mathrm{FR}} \left( \frac{\delta G}{\delta \lambda}(\lambda, \cdot)\lambda, \xi \right).$$

Since $\xi \in T_\lambda^{\mathrm{FR}}(\mathcal{M})$ is arbitrary, it follows from (41) that

$$\mathrm{grad}^{\mathrm{FR}} G(\lambda)(x) = \frac{\delta G}{\delta \lambda}(\lambda, x)\lambda(x),$$

and consequently by (42),

$$\partial_t \lambda_t(x) = -\frac{\delta G}{\delta \lambda}(\lambda_t, x)\lambda_t(x).$$

# D    Example: Convergence of discrete-time Fisher-Rao dynamics for a non-interactive min-max game

In this section, we provide an example of a game for which the convergence of the discrete-time Fisher-Rao dynamics to an MNE (measured in NI error) occurs at rate $\mathcal{O}\left(\frac{1}{\eta n}\right)$ in the non-regularized setting, and at rate $\mathcal{O}\left(e^{-\frac{1}{2}\sigma\eta n}\right)$ (to the unique MNE) when regularization is added to the objective function. Here, $\eta > 0$ and $\sigma > 0$ denote the discretization step-size and regularization parameter, respectively.

To be precise in what follows, we use the terms "discrete-time Fisher-Rao" and "Mirror Descent-Ascent" as synonyms. As it is argued in Wang & Chizat (2023) (at the end of the first bullet point on page 8) a worst-case example which shows convergence of the Fisher-Rao dynamics in discrete time in terms of the NI error at a rate $\mathcal{O}\left(\frac{1}{\eta n}\right)$ can be constructed by following the arguments from Proposition 5.5, Setting II (see the proof of Proposition 5.1 for this specific setting) in Chizat (2022).

Chizat (2022, Proposition 5.5) discusses the case of minimizing a (non-linear) convex function along the iterates of the Mirror Descent algorithm (see Algorithm 1 in (Chizat, 2022)). In order to establish the connection between the setting in Chizat (2022) and the min-max games setting, we note that removing the terms corresponding to the Wasserstein flow in the Conic Particle Mirror-Prox (CP-MP) algorithm in Wang & Chizat (2023) (see the update rule below Lemma 2.1) and taking $L = 1$ retrieves the Mirror Descent-Ascent (MDA) scheme with step-size $\eta > 0$ for min-max games in which players update simultaneously. The zero-step-size limit of the MDA scheme is precisely the Fisher-Rao gradient flow.

Following Setting II in the proof of Proposition 5.1 in Chizat (2022), we construct an example for the convergence of the Fisher-Rao dynamics in discrete time in terms of the NI error at a rate $\mathcal{O}\left(\frac{1}{\eta n}\right)$ for the min-max setting in Wang & Chizat (2023) as follows.

First, recall that the min-max game considered in Wang & Chizat (2023) is given by setting $F = \int_{\mathcal{Y}}\int_{\mathcal{X}} f(x,y)\mu(\mathrm{d}x)\nu(\mathrm{d}y)$ and $\sigma = 0$ in (1), i.e.,

$$\min_{\mu\in\mathcal{P}(\mathcal{X})}\max_{\nu\in\mathcal{P}(\mathcal{Y})} F^0(\mu,\nu), \quad \text{with } F^0(\mu,\nu) := \int_{\mathcal{Y}}\int_{\mathcal{X}} f(x,y)\mu(\mathrm{d}x)\nu(\mathrm{d}y). \tag{43}$$

Consider $f(x,y) = \Phi(x) - \Phi(y)$, where $\Phi$ is any smooth function such that $\Phi(x) = \frac{1}{2}\|x\|_2^2$ when $\|x\|_2 \leq \frac{1}{2}$ and $\Phi(x) \geq \frac{1}{4}$ otherwise. For this choice of $f$, initial measures $(\mu^0,\nu^0)$ supported on the entire space $\mathcal{X}$ and $\mathcal{Y}$, and any step-size $\eta > 0$, the simultaneous MDA update reads

$$\begin{cases} \mu^{n+1} = \arg\min_\mu\{\int \Phi(x)(\mu - \mu^n)(\mathrm{d}x) + \frac{1}{\eta}\mathrm{D}_{\mathrm{KL}}(\mu,\mu^n)\}, \\ \nu^{n+1} = \arg\max_\nu\{\int(-\Phi(y))(\nu - \nu^n)(\mathrm{d}y) - \frac{1}{\eta}\mathrm{D}_{\mathrm{KL}}(\nu,\nu^n)\}. \end{cases}$$

Hence, we can write $\mu^n(x) \propto \mu^0(x)\exp(-n\eta\frac{1}{2}\|x\|_2^2)$ and $\nu^n(y) \propto \nu^0(y)\exp(-n\eta\frac{1}{2}\|y\|_2^2)$, which are Gaussian distributions of variance $\frac{1}{\eta n}$. Therefore, for $V^0$ given by (43), we can compute

$$\mathrm{NI}(\mu^n,\nu^n) = \max_{(\mu,\nu)}\left(F^0(\nu^n,\mu) - F^0(\nu,\mu^n)\right) = \int_{\mathcal{Y}}\Phi(y)\nu^n(\mathrm{d}y) + \int_{\mathcal{X}}\Phi(x)\mu^n(\mathrm{d}x) - \min_\nu\int_{\mathcal{Y}}\Phi(y)\nu(\mathrm{d}y) - \min_\mu\int_{\mathcal{X}}\Phi(x)\mu(\mathrm{d}x).$$

Note that $\nu \mapsto \int\Phi(y)\nu(\mathrm{d}y)$ and $\mu \mapsto \int\Phi(x)\mu(\mathrm{d}x)$ have unique minimizers $\nu^* = \delta_0$ and $\mu^* = \delta_0$, respectively, and $\Phi(0) = 0$. Now, assume for example that $(\mu^0,\nu^0)$ are uniform distributions on $\mathcal{Y}$ and $\mathcal{X}$, respectively (recall that $\mathcal{X},\mathcal{Y}$ are assumed to be compact in Wang & Chizat (2023)). Then, following the proof of (Chizat, 2022, Proposition 5.5), we obtain

$$\int_{\mathcal{Y}}\Phi(y)\nu^n(\mathrm{d}y) \approx \frac{1}{2\eta n},$$

and analogously

$$\int_{\mathcal{X}}\Phi(x)\mu^n(\mathrm{d}x) \approx \frac{1}{2\eta n}.$$

Hence,

$$\mathrm{NI}(\mu^n, \nu^n) = \max_{(\mu,\nu)}(F^0(\nu^n, \mu) - F^0(\nu, \mu^n)) = \int \Phi(y)\nu^n(\mathrm{d}y) + \int \Phi(x)\mu^n(\mathrm{d}x) \approx \frac{1}{\eta n}.$$

Now, consider the entropic regularization of (43), i.e.,

$$\min_{\mu \in \mathcal{P}(\mathcal{X})} \max_{\nu \in \mathcal{P}(\mathcal{Y})} F^\sigma(\mu, \nu), \quad \text{with } F^\sigma(\mu, \nu) := \int_{\mathcal{Y}} \int_{\mathcal{X}} f(x, y)\mu(\mathrm{d}x)\nu(\mathrm{d}y) + \sigma \, \mathrm{D_{KL}}(\mu|\pi) - \sigma \, \mathrm{D_{KL}}(\nu|\rho),$$

where $\sigma > 0$ is the regularization parameter and $\pi, \rho$ are defined as in (1). Using the definition of $f$, a straightforward calculation shows that

$$F^\sigma(\mu, \nu) = F^\sigma_\pi(\mu) - F^\sigma_\rho(\nu),$$

where

$$F^\sigma_\pi(\mu) = \int_{\mathcal{X}} \Phi(x)\mu(\mathrm{d}x) + \sigma \, \mathrm{D_{KL}}(\mu|\pi),$$

$$F^\sigma_\rho(\nu) = \int_{\mathcal{Y}} \Phi(y)\nu(\mathrm{d}y) + \sigma \, \mathrm{D_{KL}}(\nu|\rho).$$

In this case, the simultaneous MDA update reads

$$\begin{cases} \mu^{n+1} = \arg\min_\mu \{\int \frac{\delta F^\sigma_\pi}{\delta\mu}(\mu^n, x)(\mu - \mu^n)(\mathrm{d}x) + \frac{1}{\eta}\mathrm{D_{KL}}(\mu|\mu^n)\}, \\ \nu^{n+1} = \arg\max_\nu \{-\int \frac{\delta F^\sigma_\rho}{\delta\nu}(\nu^n, y)(\nu - \nu^n)(\mathrm{d}y) - \frac{1}{\eta}\mathrm{D_{KL}}(\nu|\nu^n)\}. \end{cases}$$

Note that since the game is non-interactive due to the definition of $f$, we essentially need to solve two independent optimization problems. Observe that $F^\sigma_\pi$ and $F^\sigma_\rho$ are both $\sigma$-strongly convex relative to $\mathrm{D_{KL}}$ in the sense of Lemma 3.1, i.e., for any $\mu', \mu \in \mathcal{P}(\mathcal{X})$, we have

$$F^\sigma_\pi(\mu') - F^\sigma_\pi(\mu) \geq \int \frac{\delta F^\sigma_\pi}{\delta\mu}(\mu, x)(\mu' - \mu)(\mathrm{d}x) + \sigma \, \mathrm{D_{KL}}(\mu'|\mu),$$

and analogously for $F^\sigma_\rho$. Thus, we can follow the proof of (Aubin-Frankowski et al., 2022, Theorem 4) with the distinction that the Bregman divergence in Aubin-Frankowski et al. (2022) is actually the KL divergence $\mathrm{D_{KL}}$ in our case. More specifically, in our setup, equation (39) from the proof of (Aubin-Frankowski et al., 2022, Theorem 4), becomes

$$F^\sigma_\pi(\mu^{n+1}) \leq F^\sigma_\pi(\mu) - \sigma \, \mathrm{D_{KL}}(\mu|\mu^n) + \frac{1}{\eta}\mathrm{D_{KL}}(\mu|\mu^n) - \frac{1}{\eta}\mathrm{D_{KL}}(\mu|\mu^{n+1}),$$

for any $\mu \in \mathcal{P}(\mathcal{X})$. Setting $\mu^*_\sigma = \arg\min_\mu F^\sigma_\pi(\mu)$ in the inequality above, using the fact that $F^\sigma_\pi(\mu^*_\sigma) \leq F^\sigma_\pi(\mu^{n+1})$, and applying the discrete-time Gronwall's lemma with the assumption $\sigma\eta \leq 1$, we can show that

$$\mathrm{D_{KL}}(\mu^*_\sigma|\mu^n) \leq \exp(-\sigma\eta n) \, \mathrm{D_{KL}}(\mu^*_\sigma|\mu^0).$$

Analogously, we have

$$\mathrm{D_{KL}}(\nu^*_\sigma|\nu^n) \leq \exp(-\sigma\eta n) \, \mathrm{D_{KL}}(\nu^*_\sigma|\nu^0).$$

Therefore, we obtain

$$\mathrm{D_{KL}}(\mu^*_\sigma|\mu^n) + \mathrm{D_{KL}}(\nu^*_\sigma|\nu^n) \leq \exp(-\sigma\eta n)\left(\mathrm{D_{KL}}(\mu^*_\sigma|\mu^0) + \mathrm{D_{KL}}(\nu^*_\sigma|\nu^0)\right).$$

Then, as we argue in the proof of Theorem 2.3, we can consequently show that

$$\mathrm{NI}(\mu^n, \nu^n) \leq 2C_\sigma \exp\left(-\frac{1}{2}\sigma\eta n\right)\sqrt{\mathrm{D_{KL}}(\mu^*_\sigma|\mu^0) + \mathrm{D_{KL}}(\nu^*_\sigma|\nu^0)}.$$

In conclusion, in the case where there is no interaction between players, adding regularization leads to exponential convergence of the discrete-time Fisher-Rao dynamics to the unique MNE of the game, whereas no regularization guarantees only $\mathcal{O}\left(\frac{1}{\eta n}\right)$ convergence rate to an MNE.

