# OpenReview forum: "A Fisher-Rao gradient flow for entropic mean-field min-max games"
_TMLR — Accepted by TMLR_

### Review · Reviewer_sTP3 · 2024-05-30

**Summary Of Contributions:**

This work considers the Fisher-Rao gradient descent-ascent flow for finding the mixed Nash equilibria of the entropic-regularized min-max game. More precisely, regarding the mixed behavior of agents as a probability distribution over the set of strategies, the problem can be formulated as \begin{equation*} \min_{\nu \in \mathcal{P}(\mathcal{X})} \max_{\mu \in \mathcal{P}(\mathcal{Y})}V^\sigma (\nu,\mu), \qquad \text{ with } V^\sigma(\nu,\mu) = F(\nu,\mu)+\frac{\sigma^2}{2}(\mathrm{KL}(\nu|\pi) - \mathrm{KL}(\mu|\rho)).\end{equation*}
	Here $F$ is a sufficiently regular functional that is convex in $\nu$ and concave in $\mu$, $\sigma>0$ is the regularization parameter, and $\rho,\pi$ are reference measures. The gradient flow dynamics then has the form (after suitable normalization)
	\begin{equation*} \partial_t \nu_t (x)  = -\nu_t(x) \frac{\delta V^\sigma}{\delta \nu_t}(\nu_t,\mu_t,x), \qquad \partial_t \mu_t(y)  = \mu_t(y) \frac{\delta V^\sigma}{\delta \nu_t}(\nu_t,\mu_t,y).\end{equation*}
	The authors proved that, if the initial conditions $(\nu_0,\mu_0)$ are pointwisely bounded above and below by $\pi,\rho$, then for any fixed $\sigma$, the gradient flow dynamics is well-posed for all time and converges to the unique Nash equilibrium $(\nu_\sigma^\star, \mu_\sigma^\star)$ exponentially with rate $e^{-\frac{\sigma^2t}{2}}$. Finally, the Nikaidò-Isoda error also converges to zero with rate $\frac{1}{t}$.

**Audience:**

Yes

**Broader Impact Concerns:**

None.

**Claims And Evidence:**

Yes

**Requested Changes:**

1. p.2,  l.6, regarding existence and uniqueness of mixed Nash equilibria, the authors only listed Lascu et al. 2023 as reference; they should also cite Domingo-Enrich et al. 2020 here, even though their setting is slightly more restrictive (i.e. $F(\nu,\mu) = \int \int K(x,y) d\nu(x)d\mu(y)$).

2. p.2, at the very beginning of Section 1.2. Fisher-Rao metric has been considered for a very long time, as it plays an important role in information geometry. I would suggest the authors to cite the following works instead:

 * Information Geometry and Its Applications, Amari.
 * Information Geometry. Nihat Ay, Jürgen Jost, Hông Vân Lê, Lorenz Schwachhöfer.

Regarding the Benamou-Brenier formulation, besides Gallouët & Monsaingeon 2017, one could also call

 * Spherical Hellinger-Kantorovich Gradient Flows, Kondratyev & Vorotnikov.

I find citing Liu et al. 2023 here gives too much credit to their work at this specific place.

3. p.3, l.3, KL is lower semicontinuous under which topology of probability measures? I would prefer the authors to specify here.

4. p.5, Assumptions 2 and 3, please refer to Definition B.1 so that the first and second variations make sense.

5. p.5, the line between (13) and (14), I would prefer to say ``the MNE of (1) satisfies the fixed point equations $\ldots$''. Also brackets are missing around the number 1.

6. p.6, Theorem 2.2. What is the precise notion of solution of (17), in terms of function space and  time regularity? You also need initial condition to have finite KL divergence, correct?

7. p.7, the first line in the second paragraph of Remark 2.5, do you mean ``explicit'' instead of ''implicit''?

8. Bottom of p.15, the second to last inequality should be $\log \frac{\nu_t(x)}{\pi(x)} \le \log R_\nu + \frac{2}{\sigma^2}C_\nu  + 2\log R_\nu + \frac{4}{\sigma^2} C_\nu.$ Namely, besides the first term, other terms on the r.h.s. should have an extra factor $\frac{2}{\sigma^2}$. Same issue for the last inequality of p.15. Corresponding changes are needed at the top of p.16.

9. p.17, l.3, I suggest the authors to emphasize ``for __any__ fixed $x\in \mathcal{M} \ldots$"

**Strengths And Weaknesses:**

Strengths:

The mathematical results in this paper are new and correct to me. In particular, the long time convergence result is not achieved in previous works that consider the corresponding Wasserstein flows (as a comparison, the result of Lu 2023 holds only when the two dynamics have a timescale separation).

The setting of the problem is also interesting. There have been various works considering various min-max flows for finding mixed Nash equilibria, and this work is a nice addition to the field.


Weakness: It would be nice if the authors could elaborate more about the motivation for considering Fisher-Rao gradient flows. The dynamics in Domingo-Enrich already have Fisher-Rao terms, which should be mentioned at an earlier and more visible place. Also for sampling purposes, the potential ability of the dynamics overcoming metastability is the main motivation (one could modify this sentence for other purposes like training neural networks). A small discussion along these lines would be nice.

---

> ### Author Response · Authors · 2024-07-09
>
> We would like to thank the reviewer for appreciating the contribution of our paper, for their feedback and questions. We uploaded an updated version of the paper incorporating the reviewer's requested changes.
>
> We include a complete list of introduced changes:
>
> Weakness: We have added additional comments about the motivation for studying FR flows, see the second paragraph of the introduction, and the last paragraph of Subsection 1.4.
>
> 1. The reference to (Domingo-Enrich et al. 2020) has been added.
> 2. Additional references have been added at the beginning of Section 1.2.
> 3. We specified that KL is lower semi-continuous with respect to the weak convergence of probability measures.
> 4. We referred to Definition B.1 in Assumptions 2 and 3.
> 5. We rephrased the line between (13) and (14) as ``the MNE of (1) satisfies the fixed point equations''. The brackets around number 1 are added.
> 6. A comment about the notion of the solution to (17) (now (19)) has been added. Regarding the initial condition, by Assumption 4, $(\nu_0, \mu_0)$ are absolutely continuous with respect to $\pi$ and $\rho,$ respectively, which ensures that $\operatorname{D_{KL}}(\nu_0|\pi)$ and $\operatorname{D_{KL}}(\mu_0|\rho)$ are finite. Therefore, in the proof of existence of the flow we do not need to impose additional assumptions on the initial condition.
> 7. This has been corrected.
> 8. Thank you for pointing this out, we have corrected the constants.
> 9. This has been corrected.

---

> ### Comment · Reviewer_sTP3 · 2024-07-19
> **Additional questions on well-posedness**
>
> Sorry I realized that I overlooked some points when writing my previous review. If possible I would like to hear the response from the authors as soon as possible:
>
> 1. Proof of Theorem 2.2, the existence part: at the end, after constructing a Cauchy sequence and subsequently a limit, it is not automatic that $\phi(\nu_t^{(n)},\mu_t^{(n)})$ also converges to $\phi(\nu_t,\mu_t)$. In this case it is probably not so difficult to verify this by existing properties, but this does require justification. Also a smaller issue is that, to prove the iterative sequence is Cauchy sequence, it is insufficient that $\frac{t^n}{n!}<1$, but rather, it is summable.
>
> 2. More importantly, I'm not so sure how you get uniqueness of the solution. The biggest reason is that you are not using the Banach fixed point theorem as claimed: the theorem guarantees contraction to unique fixed point if $d(\phi(x),\phi(y))\le \alpha d(x,y)$ for some $\alpha \in (0,1)$ for arbitrary $x,y$, which is not what is proved in the work.
>
> 3. Could you comment on the pointwise upper and lower bounds that are assumed in your work? I imagine that lower bounds are necessary since Fisher-Rao gradient flows do have well-posedness issues at zero, but are upper bounds also necessary or you think they can be removed? Because two-sided pointwise bounds may make it not so easy for implementations.
>
> I sincerely apologize if this is too late for you, but I do hope to hear a response from you as soon as possible, ideally before I have to submit a response. Thanks a lot!

---

> > ### Author Response · Authors · 2024-07-20
> >
> > We thank the reviewer for their additional questions to which we provide answers below:
> >
> > 1. Please note that we apply the well-known observation that in order to use the Banach fixed point theorem we do not need the mapping $T$ to be a contraction everywhere, but it is sufficient if the sequence $(T(x_n))\_{n=0}^{\infty}$ is contractive only for sufficiently large $n,$ i.e., if the first inequality in the proof https://en.wikipedia.org/wiki/Banach_fixed-point_theorem
> > is satisfied only for sufficiently large $n.$ It is easy to check that the whole proof still works and one obtains a unique fixed point for the mapping $T$ (note that the continuity of $T$ along $x_n$ used in the penultimate equation in the proof is also justified by the first inequality holding only for sufficiently large $n,$ and one does not need $T$ to be a contraction everywhere for this). Hence the condition that we check is sufficient and everything else, including uniqueness, just follows from the classical theory. However, we agree that it is important that $\frac{1}{(n-1)!}\left(\frac{TC_{\text{max}}}{4}\right)^{n-1}$ is summable rather than just less than 1.
> >
> > 2. We do apply the Banach fixed point theorem as claimed, so there is no issue with uniqueness. The first inequality in the proof https://en.wikipedia.org/wiki/Banach_fixed-point_theorem is a consequence of the assumption about $T$ being a contraction, and hence holds for all $n,$ but it is clear from the proof that it does not have to hold for all $n.$ In order for the proof to work it is sufficient if it holds for sufficiently large $n.$ (see the lines starting from ``Let $\varepsilon > 0$ be arbitrary...''.). Admittedly, in our proof we do not show that our mapping $\phi$ is a contraction, but we prove exactly this weaker condition (that this first inequality from the proof holds for sufficiently large $n$), which is however sufficient to apply the Banach fixed point theorem. More specifically, using the definition of $\mathcal{TV}^{[0,T]},$ the last inequality in the proof of Lemma A.2 can be written as
> > \begin{equation}
> > \mathcal{TV}^{[0,T]} \left( (\nu_t^{(n)}, \mu_t^{(n)})\_{t \in [0,T]}, (\nu_t^{(n-1)}, \mu_t^{(n-1)})\_{t \in [0,T]} \right) \leq \left( \frac{C_{\text{max}}}{4} \right)^{n-1} \frac{T^{n-1}}{(n-1)!} \mathcal{TV}^{[0,T]} \left( (\nu_t^{(1)}, \mu_t^{(1)})\_{t \in [0,T]}, (\nu_t^{(0)}, \mu_t^{(0)})\_{t \in [0,T]} \right),
> > \end{equation}
> > where $\phi\left((\nu_t^{(n-1)}, \mu_t^{(n-1)})\_{t \in [0,T]}\right) \coloneqq (\nu_t^{(n)}, \mu_t^{(n)})\_{t \in [0,T]}.$ It suffices that this inequality holds for sufficiently large $n$ and $\frac{1}{(n-1)!}\left(\frac{TC_{\text{max}}}{4}\right)^{n-1}$ is summable (rather than just less than $1$). The latter condition follows immediately since $\sum_{n=1}^{\infty} \frac{1}{(n-1)!}\left(\frac{TC_{\text{max}}}{4}\right)^{n-1}$ is the power series of the exponential function $\exp\left(\frac{TC_{\text{max}}}{4}\right),$ which is finite for any fixed finite time horizon $T > 0.$
> >
> > 3. We need the upper bound assumption to make sure that the uniform in time bound on KL divergence in (20) holds, which is crucial for our proof of existence and differentiability. We claim that this assumption is as natural as the assumption on the lower bound - we assume that the Radon-Nikodym derivative of $\nu$ with respect to $\pi$ (and similarly for $\mu$ and $\rho$) is bounded both from below and above, which just means that the measures are "comparable" - this will be naturally satisfied in most applications just by initializing from a Gaussian, but from the proof it is evident that the closer we initialize to the MNE, the better convergence constants we get. Moreover, the assumption of having both a lower and upper bound on the ratio implies equation (18), and hence we have $\operatorname{D_{KL}}(\nu_{\sigma}^*|\nu_0) + \operatorname{D_{KL}}(\mu_{\sigma}^*|\mu_0) < \infty,$ which is essential because it prevents the right-hand side in the convergence estimates from Theorem $2.3$ to explode.

---

> ### Comment · Reviewer_sTP3 · 2024-07-20
> **Follow-up comments on well-posedness**
>
> Honestly I don't quite understand your response, but after some really careful thinking, I think well-posedness is indeed true for the dynamics, but one needs more careful argument to conclude. In particular, I find it much easier to really bound $d(T(x),T(y))$ by $d(x,y)$ and show that $T^n$ is eventually a contraction, instead of writing everything using the same iterative sequence. To be more precise:
>
> I agree that the sequence $(\nu_t^{(n)},\mu_t^{(n)})$ converges in TV norm to some $(\nu_t,\mu_t)$. However, I think one needs extra argument to show that $(\nu_t,\mu_t)$ is a fixed point. The reason is that it's not completely clear whether $T$ is continuous along the iteration. If we can prove continuity of $T$ (which is much weaker than eventual contraction), then we not only have $x_n \rightarrow x_*$ as before, but also $x_{n+1} = T(x_n) \rightarrow T(x_*)$ which shows $x_* = T(x_*)$ is a fixed point. However, I don't see precisely where continuity of $T$ is shown. In other words, I only see convergence to $x_*$, but not convergence to $T(x_*)$.
>
> The following is how I would argue this: look at the rhs of (29) (same argument for (30)) $e^{-\frac{\sigma^2}{2}t}\log \nu_0(x) -\int_0^t \frac{\sigma^2}{2}e^{-\frac{\sigma^2}{2}(t-s)}\Big( \frac{2}{\sigma^2}\frac{\partial F}{\partial \nu}(\nu_s^{(n-1)},\mu_s^{(n-1)},x ) - \log \pi(x) -D_{KL}(\nu_s^{(n-1)}|\pi)\Big)ds$; the first and third terms do not depend on $n$; the second term converges thanks to (12); while for the last term, since convergence in $TV$ actually implies strong convergence in $L^1(\mathbb{R}^d)$, one can actually use dominated convergence theorem to argue that the KL term also has to converge. From there you show your limit point is also a fixed point.
>
> As for uniqueness, the proof for the original theorem highly depends on the fact that $T$ is a contraction. That is, if you have two fixed points $x,y$, then $d(x,y) = d(T(x),T(y)) <d(x,y)$ which is impossible. In the same wikipedia page where some generalizations are listed, they would either require $T^n$ is a contraction for some $n$, or $d(T^n(x),T^n(y)) \le c_n d(x,y)$ with $\sum c_n<\infty$ and for arbitrary $x,y$, but neither of these are what is directly proved in the paper.
>
> After thinking a bit further I also come to the conclusion that uniqueness is true. Suppose we have two fixed points $(\nu_t,\mu_t)$ and $(\tilde\nu_t,\tilde\mu_t)$, then following the arguments on pages 15-16, one could actually obtain
> $D_{KL}(\nu_t|\tilde\nu_t) +D_{KL}(\tilde\nu_t|\nu_t) \lesssim TV(\nu_t,\tilde\nu_t)\int_0^t e^{-\frac{\sigma^2}{2}(t-s)} \Big(TV(\nu_s,\tilde\nu_s) + TV(\mu_s,\tilde\mu_s)\Big) ds $; combine this with the identical inequality for $\mu_t,\tilde\mu_t$ and use Pinsker's inequality, one could use Gronwall and $\nu_0=\tilde\nu_0$ to conclude that $\nu_t=\tilde\nu_t$.
>
> I hope my response clarifies the things, and if the authors agree with me, I would like to see the authors update their manuscript the way I indicated here. Thanks and please let me know if you have further questions.

---

> > ### Author Response · Authors · 2024-07-22
> > **Updated manuscript**
> >
> > We thank the reviewer for their detailed comments. After considering the reviewer's argument, we agree that the fixed point property and uniqueness do not follow immediately from our proof. Consequently, we have modified the proof of Theorem 2.2 according to the reviewer's suggestions, and have uploaded the updated manuscript.

---

### Review · Reviewer_Fcxx · 2024-06-21

**Summary Of Contributions:**

This paper explores the convergence of a Fisher-Rao (FR) gradient flow within the context of solving convex-concave min-max games, where the objective is a bivariate functional of the two input distributions, combined with KL divergence regularizations. The authors prove that  Fisher-Rao gradient flow finds the mixed (and regularized) Nash equilibrium at a sublinear rate, where the convergence is measured via the NI function. To establish this result, the authors use KL divergence as the Lyapunov function and prove that it converges at a linear rate along the course of FR gradient flow.

**Audience:**

Yes

**Broader Impact Concerns:**

No concern.

**Claims And Evidence:**

Yes

**Requested Changes:**

1. Add more discussions about the assumptions. See Point 2 above.

2. Highlight that the functional derivative is "flat derivatives" and discuss this concept in details.

3. Discuss how to implement this Fisher Rao gradient flow, especially using particle methods. Can it be implemented using particle methods? Or do we need to maintain a density?

4. Question: Is the assumption that $\int \frac{\delta V}{\delta \mu} (\mu_t, \nu_t) = 0$ without loss of generality? Can everything go through without this condition? Also, should this condition hold for all $(\mu_t, \nu_t)$?

5. It seems that the proof analysis is very similar to that of Replicator Dynamics with entropy regularization for zero sum (or even monotone) matrix games. See, e.g., "Mutation-Driven Follow the Regularized Leader for Last-Iterate Convergence
in Zero-Sum Games". I was wondering if FR gradient descent is a general version of replicator dynamics? Concretely, if distributions $\mu$, $\nu$ are supported on a finite set, and $F$ is bilinear, then your problem becomes a zero-sum matrix game. What is the corresponding algorithm and does that recover replicator dynamics? It would be great if you could add a section discussing this special case in detail. I believe this would benefit this paper a lot.

**Strengths And Weaknesses:**

Strength:
1. The convergence analysis via Lyapunov function is clear. The proof sketch provided in the introduction really benefits the paper presentation.
2. The convergence analysis seems novel.

Weakness:
1. This paper would benefit from a more in-depth derivation of Fisher Rao gradient flow. This paper introduced the variational formulation of Fisher Rao distance. However, it does not derive the formulation of Fisher Rao gradient flow, but rather directly lays out the form. It would be great to derive that form, showing why the gradient with respect to Fisher Rao metric takes that form.

2. The assumptions seem a bit strong. I understand that this paper aims to be general by considering a general functional. It would be great if some examples of functionals, beyond the bilinear objective, satisfying all these assumptions are introduced. I am especially nervous about Asssumption 3, which involves second order functional derivatives. It seems that the notion of functional derivative is different from the commonly seen Frechet derivative. I would encourage the authors also put the introduction of this version of derivative in the main paper (e.g., the background) and give a few concrete examples.

---

> ### Author Response · Authors · 2024-07-09
> **First part of the response**
>
> We would like to thank the reviewer for appreciating the contribution of our paper, for their feedback and questions. We would like to address the weaknesses and questions formulated by the reviewer:
>
> Weaknesses. (1) We thank the reviewer for the suggestion. The derivation of the Fisher-Rao gradient flow utilizes techniques from Riemannian geometry and has already been carried out, for example, in Section 2.2 in [1] and Appendix C.2.1 in [2]. In the first version of the paper, we cited [1] since their formulation of the FR gradient flow is the closest to ours. We avoided rewriting the calculations done in either [1] or [2], and instead explained (after equation (4)) the roles of the reaction term $r_t(x)$ and of the integral in (4), which is sufficient for the purposes of our paper. We uploaded a new version in which we also refer to Appendix C.2.1 in [2] (besides Section 2.2 in [1]) as an alternative source for the derivation of the FR gradient flow, see the first paragraph of Section 1.2.
>
> (2) 2.1. As we stressed in Remark 2.1, we focused on the bilinear objective function since it has been widely used in ML applications. Another example that can be seen as a particular case of our general framework is the objective function of Wasserstein-GANs with gradient penalty [3,4]. The gradient penalty regularization is added in order to ensure that the discriminator is $1$-Lipschitz along the interpolation between the generated and true data. Thus, the objective function can be expressed as
> \begin{equation}
> \min_{\nu} \max_{\mu} \int \int f(x,y) (\nu-\hat{\nu})(\mathrm{d}x)\mu(\mathrm{d}y) + t\lambda \int \int (\|\nabla_{x} f(x,y)\|-1)^2 (\nu-\hat{\nu})(\mathrm{d}x)\mu(\mathrm{d}y)\\ + (1-t)\lambda R\left(\int \int (\|\nabla_{x} f(x,y)\|-1)^2 \hat{\nu}(\mathrm{d}x)\mu(\mathrm{d}y)\right),
> \end{equation}
>
> where $x \mapsto f(x,y)$ is assumed to be a differentiable $1$-Lipschitz function, $R:\mathbb R \to \mathbb R$ is a twice differentiable concave function with bounded derivatives, $\hat{\nu}$ is the distribution of the true data, $\lambda > 0$ is a regularization parameter, and $t \in [0,1].$ Since $R$ is concave, the term inside $R$ is linear in $\mu$ and independent of $\nu,$ it follows that the third term in the optimization problem is concave in $\mu.$ This fact together with the linearity in $\nu$ and $\mu$ of the first two terms shows that the objective function satisfies Assumption $1.$ Since $f$ is differentiable and $1$-Lipschitz, and $R$ is twice differentiable with bounded derivatives, Assumptions 2 and 3 are satisfied if we also impose that $f$ is bounded. We added this example in Remark 2.1.
>
> 2.2. Regarding the differentiability on the space of probability measures, it is crucial that we work with another notion of derivative because the Frechet derivative is defined for normed spaces, and the space of probability measures is not even a vector space. The notion of ``flat derivative'' is not new in the mean-field optimization literature and has been extensively used in other works such as [5,6,7,8,9,10]. For concrete examples of functions $F$ and their flat derivative, one could see Example 2.1 in [9].

---

> ### Author Response · Authors · 2024-07-09
> **Second part of the response**
>
> Requested Changes.
> (1) and (2) Please, see Point 2 above.
>
> (3) For sampling and training neural network purposes, a particle implementation of the FR gradient flow has already been studied in [Lu et al.. Birth-death dynamics for sampling: Global convergence, approximations and their asymptotics (2022)] and [Rotskoff et al. Global convergence of neuron birth-death dynamics (2019)] (we refer to these works in the second paragraph of the Introduction). Their approach for implementing the FR gradient flow is to view it as a birth-death process with rates $a(\nu_t, \mu_t, \cdot)$ and $b(\nu_t, \mu_t, \cdot)$, respectively. Indeed, based on the signs of $a$ and $b$ at given positions $x$ and $y$, mass of $\nu$ and $\mu$ is created/destroyed along the flow. Then the FR flow can be approximated by a discrete-time interacting particle system $\{(x^i, y^i)\}_{i=1}^{N}$, where each particle has an independent exponential clock with instantaneous birth-death rates $A(x_t^i)$ and $B(y_t^i)$ corresponding to evaluating $a$ and $b$ at the empirical measures of $\nu$ and $\mu$. Due to the logarithmic non-linearity in $a$ and $b,$ one also needs to approximate the empirical measures of $\nu$ and $\mu$ using a smooth kernel function $K.$ If $A(x_t^i) > 0$, then particle $x^i$ is killed with instantaneous rate $A(x_t^i)$ and another one from the remaining population of $N-1$ particles is randomly duplicated to preserve population. If $A(x_t^i) < 0$, then particle $x^i$ is duplicated with instantaneous rate $-A(x_t^i)$ and another one from the resulting population of $N+1$ particles is randomly killed. The reversed reasoning works for $B(y_t^i).$
>
> (4) The convention we make does not cause us to lose generality. From the definition of flat derivative, we have
> \begin{equation}
> F(m')-F(m)=\int_{0}^{1}\int_{\mathcal{M}}\frac{\delta F}{\delta m}(m + \varepsilon (m'-m),x)\left(m'- m\right)(\mathrm{d}x)\mathrm{d}\varepsilon,
> \end{equation}
> for all $m, m' \in \mathcal P_p(\mathcal{M}).$ Since $\left(m'- m\right)(\mathrm{d}x)$ is a finite signed measure, it follows that $\frac{\delta F}{\delta m} + c,$ for any constant $c \in \mathbb R,$ is a flat derivative of $F.$ This implies that the flat derivative of $F$ is uniquely defined up to an additive shift. Hence, we adopt the convention that we always normalize the flat derivative so that for any probability measure $m$ on $\mathcal{X}$ we have $\int_{\mathcal{X}} \frac{\delta F}{\delta m}(m,x)m(dx) = 0.$ Alternatively, we could write the Fisher-Rao gradient flow taking into account the additive constants $-\int_{\mathcal{X}} \frac{\delta V^{\sigma}}{\delta \nu}(\nu_t, \mu_t, x) \nu_t(\mathrm{d}x)$ and $- \int_{\mathcal{Y}} \frac{\delta V^{\sigma}}{\delta \mu}(\nu_t, \mu_t, y) \mu_t(\mathrm{d}y)$ (see also equation (4)), which would give
> \begin{equation*}
>     \begin{cases}
>       \partial_t \nu_t(x) = -\left(\frac{\delta V^{\sigma}}{\delta \nu}(\nu_t, \mu_t, x)-\int_{\mathcal{X}} \frac{\delta V^{\sigma}}{\delta \nu}(\nu_t, \mu_t, x) \nu_t(\mathrm{d}x)\right)\nu_t(x),\\
>       \partial_t \mu_t(y) = \left(\frac{\delta V^{\sigma}}{\delta \mu}(\nu_t, \mu_t, y) - \int_{\mathcal{Y}} \frac{\delta V^{\sigma}}{\delta \mu}(\nu_t, \mu_t, y) \mu_t(\mathrm{d}y)\right) \mu_t(y).
>     \end{cases}
> \end{equation*}
> As we explained after equation (4), the integrals in the gradient flow guarantee that $\nu_t, \mu_t$ are probability measures for all $t \geq 0.$ Instead, assuming that
> \begin{equation*}
>     \int_{\mathcal{X}} \frac{\delta V^{\sigma}}{\delta \nu}(\nu_t, \mu_t, x) \nu_t(\mathrm{d}x) = \int_{\mathcal{Y}} \frac{\delta V^{\sigma}}{\delta \mu}(\nu_t, \mu_t, y) \mu_t(\mathrm{d}y) = 0
> \end{equation*}
> along the flow $(\nu_t, \mu_t)_{t \geq 0}$ and integrating equation (5) with respect to $\mathrm{d}x$ and $\mathrm{d}y,$ respectively, still preserves the total probability mass of $\nu_t$ and $\mu_t$ equal to $1.$
>
> (5) We thank the reviewer for pointing out this reference. For games on finite strategy spaces with matrix payoff, the FR gradient flow indeed retrieves the so-called replicator dynamics studied in evolutionary game theory. In the new version of the paper, we highlight the connection between replicator dynamics and the FR gradient flow in Remark 1.1 at the end of Section 1.2. That being said, it seems from Theorem 5.1 in [11] that the FR gradient flow and the replicator-mutant dynamics (RMD) are different due to the extra term $\mu(c_i(a_i)-\pi_i^t(a_i))$ in the latter.

---

> > ### Comment · Reviewer_Fcxx · 2024-08-12
> > **Followup comments**
> >
> > I would like to thank the authors for addressing my comments.
> >
> > (1) Details of Fisher-Rao gradient and flat derivative.
> >
> > I still believe that this paper would benefit from a more detailed explanation/exposition of these notions. Directly citing some other papers does not seem enough because it is important to make the whole paper self-contained.
> >
> > (2) Implementation via particle method.
> >
> > I honestly did not follow the authors' argument. It would be great if the authors could revise the paper and carefully develop the argument and establish a rigorous theoretical result.
> >
> > (3) WGAN-GP example.
> >
> > The authors' example seems problematic. In particular, the formulation the authors propose is different from that established in references [3,4] in the response. In WGAN, we need to maximize the discriminator, which seems the function $f$ in your formulation. It is not clear what role $\mu$ plays here.

---

> ### Author Response · Authors · 2024-07-09
> **References**
>
> References:
>
> [1] Gallouet, Monsaingeon. A JKO splitting scheme for Kantorovich-Fisher-Rao gradient flows.
>
> [2] Yan, Wang, Rigollet. Learning Gaussian Mixtures using the Wasserstein-Fisher-Rao gradient flow.
>
> [3] Gulrajani et al. Improved Training of Wasserstein GANs.
>
> [4] Petzka, Fischer, Lukovnikov. On the regularization of Wasserstein GANs
>
> [5] Hu, Ren, Siska, Szpruch. Mean-Field Langevin Dynamics and Energy Landscape of Neural Networks.
>
> [6] Cardaliaguet, Delarue, Lasry, and Lions. The master equation and the convergence problem in mean field game.
>
> [7] Liu, Majka, Szpruch. Polyak–Łojasiewicz inequality on the space of measures and convergence of mean-field birth-death processes.
>
> [8] Conforti, Kazeykina, Ren. Game on random environment, mean-field Langevin system and neural networks.
>
> [9] Kazeykina, Ren, Tan, Yang. Ergodicity of the underdamped mean-field Langevin dynamics.
>
> [10] Chizat. Mean-Field Langevin Dynamics: Exponential Convergence and Annealing.
>
> [11] Abe, Sakamoto, Iwasaki. Mutation-Driven Follow the Regularized Leader for Last-Iterate Convergence in Zero-Sum Games.

---

> ### Author Response · Authors · 2024-08-21
> **Response to reviewer**
>
> We thank the reviewer for the follow-up comments.
>
> (1) In the revised version of the paper we added Example B.2 containing concrete examples of functions and their flat derivatives. We also added a formal derivation of the Fisher-Rao gradient flow in Appendix C.
>
> (2) We would like to clarify that what we briefly described is not a proof sketch of a result but a potential algorithm for implementing the Fisher-Rao gradient flow using a particle method similar to the ones developed in [12, 13, 14]. For more details, see Algorithm 1 in [12], Algorithm 1 in [13] and Section 4.5 in [14]. What does the reviewer mean by ''establish a rigorous theoretical result''?
>
> We would like to emphasise the key point regarding the implementation of Fisher-Rao gradient flows via particles methods in our setup. Extending Algorithm 1 from either [12] or [13] to min-max games is not entirely straightforward due to multiple possible choices of the order of moves between the players. In discrete time, the players can act simultaneously or sequentially. Both players can update their moves simultaneously from iteration $k$ to iteration $k+1$ or they can move in turns, i.e., only one of the player moves from iteration $k$ to $k+1$ and the other player updates their strategy after observing the first player's $k+1$-th move. Simultaneous and sequential algorithms can have different convergence rates as demonstrated in [15]. In our future work, we will provide a detailed comparison between those different variants of the algorithm.
>
>
>
> (3) Our objective function in Remark 2.1 is a simple adaptation of the WGAN objective from [3,4]. For example, consider equation (5) from [4], i.e.,
> \begin{equation*}
>     \min_{\nu} \max_{f \in Lip1} \int f(x) (\nu-\bar{\nu})(\mathrm{d}x),
> \end{equation*}
> where $\nu$ and $\bar{\nu}$ are the generated and true measures, respectively. Since we want to model the WGAN objective as an optimization problem over the space of probability measures, we will not maximize over classes of functions. Instead, we consider the following class of parametrized functions as the class of choices for the discriminator:
> $$
>     \\{x \mapsto \mathbb E[f(Y,x)]: Law(Y) = \mu \in \mathcal{P}(\mathcal{Y})\\}.
> $$
> More precisely, we parametrize the discriminator as a two-layer neural network with parameters' distribution given by $\mu$ and activation function $f.$ For each input $x$ measured against the discrepancy between the true measure $\bar{\nu}$ and the generated measure $\nu,$ the output of the discriminator is $\mathbb E[f(Y,x)].$ Now, the maximization is taken over the measure $\mu$ of the parameters $Y$ instead of the function itself. Hence, the objective finally reads
> \begin{equation*}
>     \min_{\nu} \max_{\mu} \int \mathbb E_{Y \sim \mu}[f(Y,x)] (\nu-\bar{\nu})(\mathrm{d}x) = \min_{\nu} \max_{\mu} \int \int f(x,y) (\nu-\bar{\nu})(\mathrm{d}x) \mu(\mathrm{d}y).
> \end{equation*}
> The Lipschitzness penalty terms are inspired from [3,4] but again adapted to our specific framework. For similar ways of modelling WGANs in the mean-field regime, we refer the reviewer to Section 4.2 in [8] and Section 3.1 in [9].
>
> References:
>
> For [3,4,8,9], see the references in the previous post;
>
> [12] Lu et al. Accelerating Langevin sampling with birth-death (2019)
>
> [13] Rotskoff et al. Global convergence of neuron birth-death dynamics (2019)
>
> [14] Lu et al. Birth-death dynamics for sampling: Global convergence, approximations and their asymptotics (2022)
>
> [15] Lascu et al. Mirror Descent-Ascent for mean-field min-max problems (2024)

---

### Review · Reviewer_Eqm9 · 2024-06-27

**Summary Of Contributions:**

The paper analyzes the convergence properties of Fisher-Rao gradient flow for regularized min-max games. The existence and unicity of the flow is established using a contractive map fixed point argument. Under a warm start condition, linear convergence to the unique equilibrium is established. A bound on the duality gap of the averaged iterates is also proposed.

**Audience:**

Yes

**Broader Impact Concerns:**

The work is of a theoretical nature.

**Claims And Evidence:**

Yes

**Requested Changes:**

The two weaknesses detailed in the previous section need to be discussed in the paper.

I believe the discussion of the NI bound is critical and doable.

Improved comparisons with particle Wassertein-Fisher-Rao schemes (where the Wassertein term is diminished) will strengthen the work and justify the need for this analysis, especially since the existence of the flow is established using standard Picard iteration arguments and would thus benefit from a strong motivation.

**Strengths And Weaknesses:**

The paper is clearly written. The proofs are well detailed and easy to follow. The convergence of the continuous flows for min-max games is not immediate since the dynamics can exhibit a cycling behavior [MPP18], making the result of this paper interesting.

The main identifiable weakness of the work is in the fact that implementable particle based discretizations of Wassertein-Fisher-Rao gradient flow already exist in the litterature [Wang, Chizat 2023]. The motivation for this result therefore needs to be argued further, why is it still interesting to establish convergence of the flow for strongly monotone settings?  For instance, the authors could discuss the fact that Wang and Chizat's work requires lower bounded step sizes (see paragraph after Thm 2.2 in Wang, Chizat) and also compare with the rates obtained when $\sigma = 0$  in WC23 (i.e no Wassertein term) to establish the cost of discretization.

The second weakness is the lack of discussion around the NI bound. First, the NI bound is established without using the strong convexity, it is thus unclear if linear rates achieved for discretized particle schemes are impossible for the flow or if it simply a loose analysis. Second it is effectively vacuous as the left hand side involves a term than can be made arbitrarily large by putting mass on low probability regions of $\mu_0, \nu_0$. The authors should discuss why the quantity on the left hand side is not effectively $+\infty$ and why it remains intersting.


[MPP18] Cycles in adversarial regularized learning, Mertikopoulos, Papadimitriou, Pilouras

---

> ### Author Response · Authors · 2024-07-09
>
> We would like to thank the reviewer for appreciating the contribution of our paper, for their feedback and questions. We would like to address the weaknesses and questions formulated by the reviewer:
>
> We agree with the reviewer that the analysis of the NI bound could be improved, and we have done so in the revised version of the paper. Specifically, the second statement in Theorem 2.3 has been changed. We proved last-iterate exponential convergence of the flow with respect to NI by leveraging strong-convexity-strong-concavity of the objective $V^{\sigma}.$ When the objective function is strongly-convex-strongly-concave, the dynamics do not end up on a cycling trajectory, thus making it possible to show the last-iterate convergence. However, for convex-concave objective functions, as the reviewer observed, the dynamics manifest cycling behaviour, and only convergence of the time-averaged iterates is guaranteed.
>
> Moreover, in the strongly-convex-strongly-concave case, the fact that $\operatorname{D_{KL}}(\nu_{\sigma}^*|\nu_0) + \operatorname{D_{KL}}(\mu_{\sigma}^*|\mu_0) < \infty$ follows due to our warm start assumption, see equation (18). Indeed, since the MNE $(\nu_{\sigma}^*, \mu_{\sigma}^*)$ satisfies the fixed point equations (16) and (17), together with Assumption 2, we observe that the warm start condition, i.e., Assumption $4,$ is equivalent to equation (18). This implies that we can bound the ratio between the densities of $\nu_{\sigma}^*$ and $\nu_0$ from below and above, and analogously for $\mu_{\sigma}^*$ and $\mu_0.$ Thus, cf. (18), we have
> \begin{equation}
> \frac{1}{\hat{R}\_{\nu}} \leq \frac{\nu_{\sigma}^*(x)}{\nu_0(x)} \leq \frac{1}{\hat{r}\_{\nu}},
> \end{equation}
> for some $\hat{R}\_{\nu} > 1$ and some $\hat{r}\_{\nu} > 0.$ Taking logarithm of both sides, and integrating with respect to $\nu_{\sigma}^*(x),$ gives that
> \begin{equation}
> \log \frac{1}{\hat{R}\_{\nu}} \leq \operatorname{D_{KL}}(\nu_{\sigma}^*|\nu_0) \leq \log \frac{1}{\hat{r}\_{\nu}}.
> \end{equation}
> Analogously, one can prove a similar bound for $\operatorname{D_{KL}}(\mu_{\sigma}^*|\mu_0).$ On the contrary, as the reviewer remarked, it is not obvious how to guarantee that $\max_{\nu}\operatorname{D_{KL}}(\nu|\nu_0) + \max_{\mu}\operatorname{D_{KL}}(\mu|\mu_0)$ is finite. We believe that the result in the revised version is a significant improvement over the previous result.
>
> In the updated version of the paper, we have also extended the Related Works section (Subsection 1.4) by providing a more detailed comparison between our work and [Wang, Chizat 2023] in order to stress the motivation for studying the convergence of the continuous-time gradient flow for the regularized game.

---

> > ### Comment · Reviewer_Eqm9 · 2024-07-26
> > **Response to authors**
> >
> > I thank the authors for updating Theorem 2.3 to make it non-vacuous. It is now a better result as it actually exploits the added regularization.
> >
> > The more detailed comparison with the work of Wang and Chizat is welcome. I believe their initialization assumption in their work is essentially equivalent to your Assumption 4 so I would not insist on this difference. The writing of the motivation for studying continuous time schemes could be fleshed out a little further: what is the worst-case you refer to where linear convergence for continous time holds but the implementable discretization converges sublinearly ? I'm unsure this worst case is in the strongly monotone setting. If it were, then this strengthens your work as it helps to explicitly characterizes the cost of discretization, showing what is lost when making the algorithm implementable.

---

> ### Author Response · Authors · 2024-08-21
> **Response to reviewer I**
>
> We thank the reviewer for appreciating the improvements in the revised version. We would like to elaborate on the question raised by the reviewer.
>
> The worst-case example refers to comparing the setting of [Wang, Chizat; 2023], i.e., non-regularized min-max game with bilinear objective versus entropy regularized min-max game with bilinear objective.
>
> To be precise in what follows, we use the terms ''discrete-time Fisher-Rao'' and ''Mirror Descent-Ascent'' as synonyms.
>
> As it is argued in [Wang, Chizat; 2023] (at the end of the first bullet point on page 8) a worst-case example which shows convergence of the pure Fisher-Rao dynamics in discrete time in terms of the NI error at a rate $1/n$ can be constructed by following the arguments from Proposition 5.5, Setting II (see the proof of Proposition 5.1 for this specific setting) in [1].
>
> The work [1] (in particular, Proposition 5.5) discusses the case of minimizing a (non-linear) convex function along the iterates of the Mirror Descent algorithm (see Algorithm 1 in [1]). In order to establish the connection between [1] and the min-max setting, note that removing the terms corresponding to the Wasserstein flow in the Conic Particle Mirror-Prox (CP-MP) algorithm in [Wang, Chizat; 2023] (see the update rule below Lemma 2.1) and taking $L=1$ retrieves the Mirror Descent-Ascent (MDA) scheme with step-size $\eta > 0$ for min-max games in which players update simultaneously. The zero-step-size limit of the MDA scheme is precisely the Fisher-Rao gradient flow.
>
> Following Setting II in the proof of Proposition 5.1 in [1], we construct an example for the convergence of the pure Fisher-Rao dynamics in discrete time in terms of the NI error at a rate $1/n$ for the min-max setting in [Wang, Chizat; 2023] as follows. Consider $f(x,y) = \Phi(x) - \Phi(y),$ where $\Phi$ is any smooth function such that $\Phi(x) = \frac{1}{2}\|x\|\_2^2$ when $\|x\|\_2 \leq \frac{1}{2}$ and $\Phi(x) \geq \frac{1}{4}$ otherwise. For this choice of $f$ the simultaneous MDA update reads
> \begin{equation}
> \begin{cases}
>     \mu^{n+1} = \argmin_{\mu} \\{\int \Phi(x)(\mu-\mu^n)(\mathrm{d}x) + \frac{1}{\eta} KL(\mu, \mu^n)\\},\\
>     \nu^{n+1} = \argmax_{\nu} \\{\int (-\Phi(y))(\nu-\nu^n)(\mathrm{d}y) - \frac{1}{\eta} KL(\nu, \nu^n)\\}.
> \end{cases}
> \end{equation}
> Hence, we can write $\mu^n(x) \propto \mu^0(x) \exp(-n \eta \frac{1}{2} \|x\|\_2^2)$ and $\nu^n(y) \propto \nu^0(y)\exp(-n \eta \frac{1}{2} \|y\|\_2^2),$ which are Gaussian distributions of variance $\frac{1}{\eta n}.$ Therefore, for $F$ given by (1.1) in [Wang, Chizat; 2023], we can compute
> \begin{equation}
>     NI(\mu^n, \nu^n) = \max_{(\mu, \nu)}(F(\nu^n, \mu) - F(\nu, \mu^n)) = \int \Phi(y) \nu^n(\mathrm{d}y) + \int \Phi(x)\mu^n(\mathrm{d}x) - \min_{\nu} \int \Phi(y) \nu(\mathrm{d}y) - \min_{\mu} \int \Phi(x)\mu(\mathrm{d}x).
> \end{equation}
> Note that $\nu \mapsto \int \Phi(y) \nu(\mathrm{d}y)$ and $\mu \mapsto \int \Phi(x)\mu(\mathrm{d}x)$ have unique minimizers $\nu^* = \delta_0$ and $\mu^*= \delta_0,$ respectively, and $\Phi(0) = 0.$ Now, assume for example that $(\mu^0, \nu^0)$ are uniform distributions on $\mathcal{Y}$ and $\mathcal{X},$ respectively (recall that $\mathcal{X}, \mathcal{Y}$ are assumed to be compact in [Wang, Chizat; 2023]). Then following the proof of Prop. 5.5 in [1], we obtain
> \begin{equation}
>     \int \Phi(y) \nu^n(\mathrm{d}y) \approx \frac{1}{2\eta n},
> \end{equation}
> and analogously
> \begin{equation*}
>     \int \Phi(x)\mu^n(\mathrm{d}x) \approx \frac{1}{2\eta n}.
> \end{equation*}
> Hence,
> \begin{equation}
>     NI(\mu^n, \nu^n) = \max_{(\mu, \nu)}(F(\nu^n, \mu) - F(\nu, \mu^n)) = \int \Phi(y) \nu^n(\mathrm{d}y) + \int \Phi(x)\mu^n(\mathrm{d}x) \approx \frac{1}{\eta n}.
> \end{equation}
> Now, consider the entropy regularized game, i.e.,
> \begin{equation*}
>     \min_{\mu} \max_{\nu} F^{\sigma}(\mu, \nu),
> \end{equation*}
> with
> \begin{equation*}
>      F^{\sigma}(\mu, \nu) = \int \int f(x,y) \mu(\mathrm{d}x)\nu(\mathrm{d}y) + \sigma KL(\mu|\pi) - \sigma KL(\nu|\rho),
> \end{equation*}
> where $\sigma > 0$ is the regularization parameter and $\pi,\rho$ are defined as in the paper. Using the definition of $f,$ a straightforward calculation shows that
> \begin{equation}
>     F^{\sigma}(\mu, \nu) = F^{\sigma}\_{\pi}(\mu) - F^{\sigma}\_{\rho}(\nu),
> \end{equation}
> where
> \begin{equation}
>     F^{\sigma}\_{\pi}(\mu) = \int \Phi(x) \mu(\mathrm{d}x) + \sigma KL(\mu|\pi),
> \end{equation}
> \begin{equation}
>     F^{\sigma}\_{\rho}(\nu) = \int \Phi(y) \nu(\mathrm{d}y) + \sigma KL(\nu|\rho).
> \end{equation}

---

> ### Author Response · Authors · 2024-08-21
> **Response to reviewer II**
>
> Observe that $F^{\sigma}\_{\pi}$ and $F^{\sigma}\_{\rho}$ are both $\sigma$-strongly convex with respect to KL divergence, i.e.,
> \begin{equation}
>     F^{\sigma}\_{\pi}(\mu') - F^{\sigma}\_{\pi}(\mu) \geq \int \frac{\delta F^{\sigma}\_{\pi}}{\delta \mu}(\mu,x)(\mu'-\mu)(\mathrm{d}x) + \sigma KL(\mu'|\mu),
> \end{equation}
> for any $\mu', \mu,$ and analogously for $F^{\sigma}\_{\rho}.$ In this case the simultaneous MDA update reads
> \begin{equation}
> \begin{cases}
>     \mu^{n+1} = \argmin_{\mu} \\{\int \frac{\delta F^{\sigma}\_{\pi}}{\delta \mu}(\mu^n,x)(\mu-\mu^n)(\mathrm{d}x) + \frac{1}{\eta} KL(\mu| \mu^n)\\},\\
>     \nu^{n+1} = \argmax_{\nu} \\{-\int \frac{\delta F^{\sigma}\_{\rho}}{\delta \nu}(\nu^n,y)(\nu-\nu^n)(\mathrm{d}y) - \frac{1}{\eta} KL(\nu| \nu^n)\\}.
> \end{cases}
> \end{equation}
> Note that since the game is non-interactive due to the definition of $f,$ we essentially need to solve two independent optimization problems. We follow the proof of [Theorem 4; 2] with the difference that the Bregman divergence in [2] is actually the KL divergence in our case. More specifically, equation (39) in the proof of [Theorem 4; 2], in our setup, becomes
> \begin{equation}
>     F^{\sigma}\_{\pi}(\mu^{n+1}) \leq F^{\sigma}\_{\pi}(\mu) - \sigma KL(\mu|\mu^n) + \frac{1}{\eta}KL(\mu|\mu^n) - \frac{1}{\eta}KL(\mu|\mu^{n+1}),
> \end{equation}
> for any $\mu.$ Note that $l=\sigma$ and $L=\frac{1}{\eta}$ in our case. Setting $\mu_{\sigma}^* = \argmin_{\mu} F^{\sigma}\_{\pi}(\mu)$ in the inequality above, using the fact that $F^{\sigma}\_{\pi}(\mu_{\sigma}^*) \leq F^{\sigma}\_{\pi}(\mu^{n+1}),$ and applying the discrete Gronwall's lemma with the assumption $\sigma \eta \leq 1,$ we can show that
> \begin{equation}
>     KL(\mu_{\sigma}^*| \mu^n) \leq \exp(-\sigma \eta n) KL(\mu_{\sigma}^*| \mu^0).
> \end{equation}
> Analogously, we can show that
> \begin{equation}
>     KL(\nu\_{\sigma}^*| \nu^n) \leq \exp(-\sigma \eta n) KL(\nu\_{\sigma}^*| \nu^0).
> \end{equation}
> Therefore, we obtain
> \begin{equation*}
>     KL(\mu_{\sigma}^*| \mu^n) + KL(\nu_{\sigma}^*| \nu^n) \leq \exp(-\sigma \eta n) (KL(\mu_{\sigma}^*| \mu^0) + KL(\nu_{\sigma}^*| \nu^0)).
> \end{equation*}
> Then as we argue in the proof of Theorem 2.3, we can consequently show that
> \begin{equation*}
>     NI(\mu^n,\nu^n) \leq 2C_{\sigma} \exp(-\frac{1}{2}\sigma \eta n) \sqrt{KL(\mu_{\sigma}^*| \mu^0) +  KL(\nu_{\sigma}^*| \nu^0)}.
> \end{equation*}
> In conclusion, in this worst-case example (where there is no interaction between players; see the definition of $f$) adding regularization leads to exponential convergence of the discrete-time Fisher-Rao flow, whereas no regularization guarantees only $1/n$ convergence rate.
>
> References:
>
> [1] Convergence Rates of Gradient Methods for Convex Optimization in the Space of Measures - https://arxiv.org/pdf/2105.08368
>
> [2] Mirror Descent with Relative Smoothness in Measure Spaces, with application to Sinkhorn and EM - https://arxiv.org/pdf/2206.08873

---

### Decision · Action_Editor_KXkD · 2024-08-17

**Recommendation:** Accept with minor revision

**Comment:**

We had good reviews and rebuttals and dialogue (and paper revisions). In the end, the reviewers were not unanimous in their recommendations but they were all close enough, with enough of a lean toward acceptance. In particular, we found that the paper meets the criteria for TMLR: enough interest, and accurate and convincing evidence.  So I'm pleased to recommend acceptance.

However, the reviewers still had a few "wishlist" items.  I think it is in the authors' best interest to address these issues, since they will make it a stronger paper and increase its impact.  In particular,

Reviewer `Fcxx` mentioned requested changes:
- a better introduction of the technical ingredients such as Fisher-Rao gradient and Flat derivative
- a better motivation of the problem setting and assumptions
- a clear articulation of the technical novelty of this paper.
- "implementation via particle method" was hard to follow, and it would be nice to develop the argument and make it more rigorous
- Fixing the issues with the WGAN-GP example (see their comments for details)

Reviewer `Eqm9` mentions fleshing out more about the motivation for continuous time (see their comments).

**Audience:**

This is a specialized paper that doesn't have the broadest appeal, but it does have sufficient appeal, in particular there are enough specialists interested in these kinds of problems and convergence results.

**Claims And Evidence:**

This is a theoretical paper, so the claims and evidence are in the form of theorems and proof.  We had 3 very competent and engaged reviewers who looked carefully at the math, and through some discussions with the authors, have come to the point where the theorems seem non-vacuous and the proofs most likely correct.

---

> ### Author Response · Authors · 2024-08-28
> **Response to Action Editor**
>
> We would like to thank the editor for organizing this high-quality review process. Regarding the items mentioned by reviewers Fcxx and Eqm9, these were addressed in responses submitted to each reviewer on the 21st August as part of the discussions. The responses were also incorporated in the (de-anonymized) camera ready version of the paper, which is now uploaded.